

# Paleomagnetic constraints on the timing and distribution of Cenozoic rotations in Central and Eastern Anatolia

Derya Gürer[1], Douwe J.J. van Hinsbergen[1], Murat Özkaptan[2], Iverna Creton[1], Mathijs R. Koymans[1], Antonio Cascella[3], Cornelis G. Langereis[1]

[1]Department of Earth Sciences, University of Utrecht, Utrecht 3584 CD, The Netherlands
[2]Department of Geophysical Engineering, Karadeniz Technical University, Trabzon 61080, Turkey
[3]Istituto Nazionale di Geofisica e Vulcanologia (INGV), Pisa 56126, Italy

*Correspondence to*: D. Gürer (derya.guerer@gmail.com)

**Abstract.** To quantitatively reconstruct the kinematic evolution of Central and Eastern Anatolia within the framework of Neotethyan subduction accommodating Africa-Eurasia convergence, we paleomagnetically assess timing and amount of vertical axis rotations across the Ulukışla and Sivas regions. We show paleomagnetic results from ~30 localities identifying a coherent rotation of a block - comprising the southern Kırşehir Block, the Ulukışla basin, the Central and Eastern Taurides, and the southern part of the Sivas basin. This block experienced a ~30° counter-clockwise vertical axis rotation since Oligocene time. Sediments in the northern Sivas region show clockwise rotations. We use the rotation patterns together with known fault zones to argue that the counter-clockwise rotating domain of south-central Anatolia was bounded by the Savcılı Thrust Zone and Deliler-Tecer Fault Zone in the north and by the African-Arabian trench in the south, the western boundary of which is poorly constrained and requires future study. Our new paleomagnetic constraints provide a key ingredient for future kinematic restorations of the Anatolian tectonic collage.

## 1 Introduction

Convergence between the African/Arabian and Eurasian plates led in the eastern Mediterranean region to the formation of the complex, and highly non-cylindrical Anatolian orogen. This non-cylindricity is expressed as the lateral appearance or disappearance of major, continent-derived deformed blocks in the orogen, whereby the prominent blocks in western and Central Anatolia (the Tavşanlı and Kırşehir zones, **Fig. 1**) disappear in eastern Anatolia. The tectonic amalgamation of these continental blocks is thought to have occurred at two or more subduction zones since Cretaceous time. At a southern subduction system, oceanic lithosphere of a former overriding plate was obducted over continental lithosphere of a Gondwana-derived, Anatolide-Tauride continental fragment in the late Cretaceous leading to deformation and metamorphism (Barrier and Vrielynck, 2008; Boztuğ et al., 2009; Menant et al., 2016; Moix et al., 2008; Parlak et al., 2012; Robertson, 2002; Robertson et al., 2009; van Hinsbergen et al., 2016). Another subduction zone existed along the northern subduction system below Eurasian lithosphere, where the obduction-related orogen collided since latest Cretaceous to Paleogene time upon closure of the Neotethys Ocean (Kaymakci et al., 2009). These collisions were shown to have been associated with major regional vertical axis block rotations throughout the Anatolian orogen (Çinku et al., 2016, 2011, Kissel



et al., 2003, 1993; Lefebvre et al., 2013; Meijers et al., 2010; Piper et al., 2010). Cenozoic rotations appear to be centered around the Kırşehir Massif of Central Anatolia (Cinku et al., 2016), suggesting that the highly non-cylindrical deformation of Anatolia may have been governed by paleogeographic distribution of continental and oceanic crust on the downgoing, African-Arabian plate. Such possible paleogeographic control on deformation and rotations may be illustrated by the

northern continental margin of Arabia that extended farther north than that of Africa, leading to Arabian collision and indentation in eastern Turkey in Late Miocene time, while subduction continues until today below western and central Turkey. This paleogeographically controlled collision distribution is thought to have generated westward extrusion of Anatolia and associated counter-clockwise rotation of Anatolia (e.g. Faccenna et al., 2006; Hüsing et al., 2009; van Hinsbergen et al., 2010; Okay and Whitney, 2010; Biryol et al., 2011). Kinematic restoration of the rotation and deformation

history of Anatolia allows putting first-order boundary conditions on the geodynamic evolution of Anatolian subduction and orogenesis. For such restorations, it is key to identify the size of coherent tectonic domains, the amount and timing of their rotation, and the structures that accommodated differential rotations between domains.

The Anatolian orogen comprises of three major continent-derived deformed blocks: the Pontides, the Kırşehir Block and the Taurides. These blocks, which were once separated by (oceanic) basins, were bent by vertical axis rotations into oroclines. In

northern Turkey, the Pontides (**Fig. 1**) comprise a Paleozoic crystalline basement derived from microcontinental crust that collided with Eurasia in the early-mid Mesozoic or before, overlain by a Mesozoic and younger volcano-sedimentary cover (Dokuz et al., 2017; Nikishin et al., 2015; Şengör and Yılmaz, 1981; Ustaömer and Robertson, 2010, 1997). To the south, the Pontides are bordered by the Izmir-Ankara-Erzincan suture zone (IAESZ), where oceanic lithosphere of the Neotethys Ocean was consumed (**Fig. 1**; Şengör and Yılmaz, 1981). This suture forms the root zone of a wide belt of ophiolites thrust

to the south on the continent-derived Kırşehir Block of Central Anatolia and the Tauride block of southern Turkey (including the Menderes massif of western Turkey), with progressively younger ages of accretion, metamorphism and exhumation from north to south (Göncüoglu, 1997; Gürer et al., 2016; Okay and Tüysüz, 1999; van Hinsbergen et al., 2016).

Paleomagnetic research has revealed that each of these major continent-derived units experienced significant rotations. The Pontides were deformed in Paleogene time into a northward convex orocline (Meijers et al., 2010; Kaymakci et al., 2003;

Cinku et al., 2011; Lucifora et al., 2016). Deformation was accommodated along thrusts associated with the IAESZ in the south, and thrusts inverting the southern Black Sea margin in the north (Kaymakci et al., 2009; Espurt et al., 2014) (**Fig. 1**). To the south, the Kırşehir block broke into three rotating blocks: the northeastern Akdag-Yozgat block (AYB), the central Kırşehir-Kırıkkale block (KKB), and the southern Avanos- Ağacören block (AAB) as shown by paleomagnetic data from Upper Cretaceous granitoids (Lefebvre et al., 2013). These rotating blocks were bordered by transpressional fault zones

active between the Late Eocene and Early Miocene (e.g. Gülyüz et al., 2013; Advokaat et al., 2014; Isik et al., 2014). It has not yet been identified where the AAB ends in the south. The Taurides, finally, were bent south of the Kırşehir block, with varying clockwise rotations documented in the Central Taurides clustering around 40°clockwise (cw) (Çinku et al., 2016; Kissel et al., 1993; Meijers et al., 2011; John D A Piper et al., 2002) and counter-clockwise (ccw) rotations in the Eastern Taurides (Çinku et al., 2016; Kissel et al., 2003) (**Fig. 1**). It remains unknown which structures accommodated the Tauride



rotations. Particularly the counter-clockwise rotations in the east are currently only constrained to post-Mesozoic (Çinku et al., 2016), probably post-middle Eocene (Kissel et al., 2003; Cinku et al., 2016).

In this paper, we aim to constrain the dimension of the counter-clockwise rotating domain(s) in central and eastern Anatolia, the timing of their rotation, and the structures that may have accommodated these rotations relative to surrounding blocks.

To this end, we collected new paleomagnetic data from the Ulukışla basin, between the Avanos-Ağacören block of the Kırşehir Block to the north and the Taurides to the south, and we test whether the Taurides and Avanos-Ağacören block underwent a coherent rotation. In addition, we collected samples from sediments of the Sivas basin, north of the eastern Taurides and covering the suture with the Pontides and AYB to identify where the northern boundary of the Tauride rotating domain is located.

## 10  2 Geological setting

### 2.1 Basement units

The Pontides mountain belt in northern Turkey is interpreted as a Gondwana-derived fragment which collided with Eurasia either in the Paleozoic (e.g., Okay and Nikishin, 2015) or in the Mid-Jurassic (e.g., Şengör and Yılmaz, 1981; Ustaömer and Robertson, 1997; Ustaömer and Robertson, 2010; Dokuz et al., 2017). The oldest radiolarian cherts in the Izmir-Ankara

mélange date back to the Middle Triassic, which is generally interpreted as the time of onset of opening of the Neotethys ocean between the Kırşehir-Tauride blocks in the south and the Pontides in the north (Tekin et al., 2002; Tekin and Göncüoglu, 2007). Below the Pontides, a northward-dipping subduction zone consuming the Neotethys ocean was active since Jurassic time (e.g., Okay et al., 2014). Closure of this ocean formed the roughly E-W trending Izmir-Ankara-Erzincan (IAESZ) suture zone which separates the Pontides from the Kırşehir-Tauride orogen (Şengör and Yılmaz, 1981).

The Kırşehir-Tauride orogen developed in Late Cretaceous time when ophiolites with supra-subduction zone geochemical affinities started forming within the Neotethys ocean around 95-90 Ma (Dilek et al., 1999; Parlak et al., 2002; Robertson et al., 2009; Yaliniz et al., 1996). This event is widely interpreted to reflect the formation of a new, intra-oceanic subduction zone within the Neotethys ocean (Dilek et al., 1999; Dilek and Furnes, 2011; Robertson et al., 2009; van Hinsbergen et al., 2015). Within 10 Myr after subduction initiation, the Kırşehir block of Central Anatolia, and the Tavşanlı zone in western

Turkey, as well as continental rocks found in isolated occurrence on the east Anatolian high plateau (Topuz et al., 2017; Yilmaz et al., 2010), arrived in the young trench below overriding plate oceanic crust – conceptually termed the Anadolu plate(s) (Gürer et al., 2016) and metamorphosed and accreted around 85-80 Ma (Boztuğ et al., 2009; Plunder et al., 2013; van Hinsbergen et al., 2016). Relics of the Anadolu plate(s) are now found as ophiolites that form the highest structural unit and are widespread in central and southern Anatolia (**Fig. 1**). Around 70-65 Ma the continental margin of the Tauride block

arrived in this trench, were subjected to HP metamorphism and accreted forming the Afyon zone (Özdamar et al., 2013; Pourteau et al., 2013). This was followed by the accretion of the Tauride block itself forming a fold-and-thrust belt consisting of Paleozoic to Eocene platform carbonates. Accretion took place from the Paleocene until at least the Late



Eocene (Demirtasli et al., 1984; Gutnic et al., 1979; Monod, 1977; Özgül, 1984), in places intruded by Eocene granitoids (Parlak et al., 2013). The Taurides mostly escaped metamorphism and accreted while the Kırşehir Block and Afyon zone were exhumed by extension in Late Cretaceous to Early Eocene time (Gautier et al., 2008, 2002; Gürer et al., 2017; Isik, 2009; Isik et al., 2008; Lefebvre et al., 2015, 2011).

The Kırşehir-Tauride orogen and overlying ophiolites collided with the Pontides in Central Anatolia in the latest Cretaceous to Paleocene, forming the IAESZ (Kaymakci et al., 2009). Subduction is still active today below the western and central parts of the Taurides, consuming oceanic crust in the west (e.g., Granot, 2016) and continental crust in the central part, south of Cyprus (e.g. Robertson et al., 2012). In the east, Arabia collided with the Eastern Taurides in Late Miocene time upon consumption of the oceanic lithosphere that once separated Arabia from the Taurides (Hüsing et al., 2009; Okay et al., 2010;

Sengör et al., 2003).

Following collision at the IAESZ, the Pontides as well as the Kırşehir-Tauride blocks underwent renewed, out-of-sequence shortening. This started with formation of the Pontides orocline in the Paleogene (Meijers et al., 2010). Further shortening and vertical axis rotations moved to the south (Çankırı basin) and continued until the Early Miocene (Kaymakci et al., 2003; 2009; Lucifora et al., 2013; Espurt et al., 2014). East of the orocline and north of the Sivas basin, the Pontides did not

experience significant rotation since the Eocene (Meijers et al., 2010a and references therein).

The rotating blocks of the Kırşehir massif were separated by the Savcılı Thrust Zone (STZ) between the KKB and AAB - a sinistral structure that underwent contraction between ~40 Ma and at least ~23 Ma (Isik et al., 2014; Advokaat et al., 2014) and the Delice-Kozakli Fault Zone (DKFZ) between the AYB and KKB that was postulated to be a dextral, transpressional structure with an estimated offset of up to 90 km (Lefebvre et al., 2013). To the west, the Kırşehir Block is bordered by the

Tuzgölü Fault Zone (TGFZ), which contains a large normal fault displacement of at least Eocene and younger age (Çemen et al., 1999).

The Taurides to the south are bent into an orocline, with clockwise rotations in the western Central Taurides (Kissel et al., 1993; Meijers et al., 2011; Çinku et al., 2016; Piper et al., 2002) and counter-clockwise rotations in the Eastern Taurides (Çinku et al., 2016; Kissel et al., 2003). The Eastern Taurides are cut by NNE-SSW trending strike slip faults. The most

western of these is the Ecemiş Fault Zone (EFZ; **Fig. 1**). It was originally defined by Yetis, (1984) and subsequently subject of numerous studies (Higgins et al., 2015; Jaffey and Robertson, 2005, 2001; Koçyiğit and Beyhan, 1998; Sarıkaya et al., 2015; Westaway et al., 2002). It is a major fault separating the Kırsehir block and Central Taurides to the west, and the Eastern Taurides to the east. It also left-laterally disrupts the connection between the Ulukışla and Sivas basins from Eocene-Oligocene time onward (Gürer et al., 2016). To the south, Lower Miocene sediments in the Adana basin seal the strike-slip

component of the Ecemis Fault Zone, which after Early Miocene time only experienced E-W directed normal faulting with only a minor strike-slip component (Alan et al., 2011; Higgins et al., 2015; Jaffey and Robertson, 2001; Sarıkaya et al., 2015).



Farther to the east, the Malatya and Ovacık Fault Zones display some 30 km of left-lateral displacement between 5 and 3 Ma (Westaway and Arger, 2001), perhaps reactivating Early Miocene normal faults (Kaymakci et al., 2006). Other sinistral faults with a reverse component include the Sariz and Gürün Fault (Kaymakci et al., 2010). Finally, along the suture of Arabia and the Taurides, the East Anatolian Fault Zone (EAFZ) has been active as a left-lateral strike-slip fault since

Pliocene time (e.g. Kaymakci et al., 2010). Together with the major right-lateral North Anatolian Fault Zone (NAFZ) that roughly follows the Izmir-Ankara suture zone, the EAFZ accommodates west-ward escape of the Anatolian orogenic collage, thought to relate to Arabian indentation (e.g. Dewey and Şengör, 1979).

### 2.2 Sedimentary basins flanking the northern Taurides

Several sedimentary basins developed on top of the sutures and recorded the geological evolution of the Anatolian orogen

from the Late Cretaceous to present. Broad similarities in their stratigraphy suggest that the Upper Cretaceous-Paleogene stratigraphic record of the Ulukışla and Sivas basins once formed a contiguous basin. Both overlie the Cretaceous ophiolites and contain stratigraphies that overlap in time with thrusting, metamorphism, exhumation, and accretion of the underlying Afyon zone and Tauride units (Akyuz et al., 2013; Clark and Robertson, 2005, 2002; Gürer et al., 2016; Poisson et al., 1996).The Ulukışla and Sivas basins became separated due to displacement along the Ecemis Fault Zone in the Late Eocene-

Oligocene (**Figs. 1-3**).

The Ulukışla basin (**Fig. 2**) contains a discontinuous stratigraphic record of laterally variable series of shallow and deeper marine clastic and carbonate sediments interlayered with volcanic rocks, overlain by continental coarse clastic rocks. The deposits range from latest Cretaceous to Miocene in age and unconformably overlie ophiolitic basement. Eocene and younger rocks of the basin also unconformably overlie metamorphic rocks of the Kırşehir block and the Afyon zone. In Late

Cretaceous time, the basin developed above the Alihoca ophiolites, during the burial of the Afyon zone below these ophiolites. From this context, the basin is interpreted as a forearc basin. During extensional exhumation of the Afyon zone along the top-to-the-north Ivriz detachment, the basin underwent N-S extension in the detachment's hanging wall and underwent widespread marine clastic deposition (Gürer et al., 2016; 2017). At the northern basin margin and close to the contact with the Kırşehir Block, a series of large-offset listric normal faults compatible with E-W extension (in present-day

coordinates). These offset sediments and the base of Paleocene volcanics indicate that E-W extension occurred simultaneously with N-S extension in the south, which prevailed until at least 56 Ma (Gürer et al., 2016). Following extension and exhumation, the Late Eocene-Oligocene history of the basin involved N-S shortening in especially the southern part of the basin, resulting in the formation of a north-verging fold with a widely exposed subvertical to overturned northern limb that marks the southern exposures of the Ulukışla basin, and subordinate top-to-the-south thrusting in the

synclinal hinge zone. Westward, these folds become more open and structures become gradually NE-SW oriented, while several anticlines and synclines deform the basal stratigraphy of the basin. The Oligocene redbeds in the Aktoprak syncline (Meijers et al., 2016) are interpreted as having been deposited during folding (Gürer et al., 2016). To the east, the folded



Ulukışla basin sediments curve towards NE-SW strikes interpreted to reflect drag folding against the major left-lateral EFZ (Gürer et al., 2016).

The Sivas region (**Fig. 3**) hosts a depocenter which is bound by the Pontides to the north, the Kırşehir block/EFZ in the west, and the Taurides in the south. Its eastern boundary is diffuse. To the south, on the northern flanks of the Tauride fold-thrust

belt, ophiolites are unconformably overlain by uppermost Cretaceous to Eocene, dominantly marine carbonate and clastic sediments and Paleocene volcanic and volcaniclastic rocks. These sediments were affected by north-verging Paleocene-Eocene thrusting **(**Cater 1991; Poisson et al., 1996; Yilmaz and Yilmaz 2006), possibly related to so far structurally poorly reconstructed thrusting in the Tauride fold-and-thrust belt underneath. In the Late Eocene-Oligocene, sedimentation became terrestrial to lacustrine. This overall sequence shows first-order similarities with the Ulukışla basin.

The younger stratigraphy, however, is significantly different, and is located to the north of a large fault zone that runs through the center of the Sivas basin. This fault zone consists of a series of thrusts and towards the west connects to the EFZ through NE-SW striking sinistral strike-slip faults, including the Kızılırmak, Gemerek-Şarkışla, Deliler, and Tecer faults (Higgins et al., 2015; **Fig. 3**). To the east, the Deliler Fault connects to the Tecer fault, and defines a N60°-70°E-trending fault with a left-lateral strike slip-component (Yilmaz & Yilmaz, 2006; Akyuz et al., 2013) that thrusts ophiolites and

Cretaceous and younger sediments over folded Oligocene redbeds to the south, all along the Sivas basin. This structure is hereafter referred to as Deliler-Tecer Fault Zone (DTFZ).

In the hanging wall of the DTFZ, the northern Sivas basin is an elongated depocenter that comprises Oligocene to Pliocene continental and marine strata including widespread Miocene evaporites overlain by continental redbeds. Salt mobility led to strong local deformation and the formation of mini-basins (Callot et al., 2014; Kergaravat et al., 2016; Pichat et al., 2016;

Poisson et al., 2016; Ribes et al., 2015). In the west, Upper Miocene to recent volcanic rocks are found close to the EFZ and its connection to the Sivas fold-and-thrust belt (Cater et al., 1991; Poisson et al., 1996; Yilmaz and Yilmaz, 2006). The subhorizontal Pliocene covers parts of the Sivas basin on both sides of the DTFZ.

### 3 Methods

#### 3.1 Sampling

An extensive sample set was obtained from 21 localities, comprising of a total of 121 sites in rocks of common age from structurally coherent regions, totalling 2118 oriented paleomagnetic cores across the Upper Cretaceous to Miocene stratigraphy of the Ulukışla and Sivas basins and sediments overlying the Tauride fold-and-thrust belt (**Fig. 2, table 1,** GPS locations of all sites are supplied in the **online appendix**). All samples were collected from sedimentary rocks. Age constraints for the sampled units come from geological maps and accompanying explanatory notes (MTA, 2002). Ages were

complemented – where available – by published biostratigraphic literature (Blumenthal, 1956; Oktay, 1973; Oktay, 1982; Demirtasli et al., 1984; Atabey et al., 1990; Clark and Robertson, 2005; Gürer et al., 2016; details supplied in the appendix).



In a few cases, we obtained new nannofossil biostratigraphy (**online appendix**). Paleomagnetic samples were collected from marine sediments (limestones, marls, and turbiditic sandstones) of Late Cretaceous to Middle Eocene as well as Miocene age, and continental clastic rocks of Oligocene and Miocene age. The samples were collected by drilling standard cores (25 mm Ø) with a gasoline-powered, water-cooled drill. Cores were oriented using a magnetic compass, and drilling orientation

as well as bedding plane were corrected for the local declination of 5°E.

### 3.2 Rock magnetism

Magnetic carriers of the NRM were determined by thermomagnetic runs in air, using a modified horizontal translation type Curie balance with a sensitivity of ~5 x $10^{-9}$ $m^2$ (Mullender et al., 1993). Depending on the expected magnetic intensity of the sample, 30-100 mg of rock powder per site was measured in a quartz vial, in a number of heating and cooling cycles

(10°C/min) to up to 700°C. Heating and cooling rates were 10°C/min. For each of the 121 sites one thermomagnetic experiment was performed.

### 3.3 Demagnetization

Samples were subjected to progressive stepwise demagnetization using either thermal (TH) or alternating field (AF) steps, from room temperature up to a maximum of 680°C, and a maximum field of 100mT respectively. To more efficiently

separate secondary and primary components by AF demagnetization, specimens were heated to 150°C to remove possible viscous or present-day field overprints caused by weathering, and to reduce the coercivities of the secondary overprint in the natural remanent magnetization (NRM) (Van Velzen and Zijderveld, 1995). Temperatures ranging 20-680°C with increments of 20–50°C were applied to thermally demagnetize the samples in a shielded ASC TD48-SC oven. The NRM after each step was measured on a 2G Enterprises horizontal 2G DC SQUID cryogenic magnetometer (noise level 3

$\times 10^{-12} Am^2$). Demagnetization steps ranging 5–100 mT with field steps of 3-10 mT were applied for AF demagnetization. AF demagnetization was performed on an in-house developed, robot assisted and fully automated 2G DC SQUID cryogenic magnetometer (Mullender et al., 2016).

### 3.4 Directional interpretation and statistical treatment

Paleomagnetic interpretation and statistical analysis were carried out using the platform independent portal

Paleomagnetism.org (Koymans et al., 2016). All data and interpretations related to this study are provided in file formats that can be imported into the portal. Stepwise demagnetization of the NRM is displayed in orthogonal vector diagrams (Zijderveld, 1967). Characteristic remanent magnetization (ChRM) directions were interpreted using principal components analysis (Kirschvink, 1980). Great circle analysis following (McFadden and McElhinny, 1988) was performed if two components were not entirely resolved upon demagnetization, or became viscous or spurious at higher temperatures or

coercivities. Mean directions were determined using Fisher (1953) statistics applied on virtual geomagnetic poles (VGPs) and errors in declination ($\Delta D_x$) and inclination ($\Delta I_x$) were calculated following Butler (1992). We applied the reliability





criteria of Deenen et al. (2011) by determining $A_{95}$ of the VGP distribution, and calculate the n-dependent values of $A_{95min}$ and $A_{95max}$: values plotting within this envelope can be explained by paleosecular variation (PSV) whilst values outside the envelope may indicate sources of enhanced scatter ($A_{95} > A_{95max}$), or underrepresentation of PSV ($A_{95} < A_{95min}$) (Deenen et al., 2014, 2011). To test the primary origin of the ChRM, field tests (reversal test, fold test) were performed where possible on

rocks of similar age (Tauxe, 2010). A fixed 45° cut-off was applied to the VGP distributions on site and locality level following (Johnson et al., 2008). Using these tests, we establish whether the locality carries a primary or secondary magnetization. In the discussion section, we then evaluate the tectonic implications of these paleomagnetic directions.

**3.5 Compilation of previous paleomagnetic results**

We have compiled previously published paleomagnetic data from the Taurides, Kırşehir Block, and Pontides previously used

to infer rotation patterns in Central Anatolia. This paleomagnetic database is provided in the Supplementary Information and in a paleomagnetism.org-compatible format. Literature references are provided in the description of the Appendix. All sites are given in tectonic coordinates (i.e., corrected for bedding tilt) where possible. All paleomagnetic directions were converted to normal polarity. The database was built following selection criteria listed in (Li et al., 2017). Because the paleomagnetic community does not normally publish their original data, but only statistical descriptions of the data set, we

have created parametrically sampled data sets for each site so as to facilitate averaging all directions rather than averages of averages (see Deenen et al., 2011). The average directions in the database are based on these parametrically sampled data sets and may differ somewhat from the published average directions. Locality averages for the segments of the Pontide orocline, the three Kırşehir Block, and the Eastern Taurides based on these literature data are provided in Table 1.

**4 Results**

**4.1 Rock magnetism**

Most samples showed decreasing magnetization up to temperatures of 550-580°C, indicative for (Ti-poor) magnetite. Often, oxidation of pyrite upon thermal demagnetization forms magnetite around 400°C (Passier et al., 2001) leading to spurious demagnetization behaviour at higher temperatures. Occasionally, the presence of hematite is seen by the removal of the magnetization at temperatures as high as 680°C. These results are consistent with Curie Balance measurements, of which

nine representative results are shown in **Fig. 4**. The following patterns were observed in thermomagnetic curves, which aided our demagnetization strategy, whereby hematite-bearing samples were preferably TH-demagnetized and pyrite bearing samples were AF-demagnetized. Near continuous, non-reversible alteration with increasing temperature was recorded by some light-red colored limestones and continental red siltstones or beige sandstones, showing an inflection around 350°C interpreted as inversion of maghemite into hematite (Dankers and others, 1978) and finally to magnetite at ∼580°C (**Fig.**

**4b,h**). The presence of hematite in some red siltstones is evident from a residual magnetic signal up to 680°C (**Fig. 4e, e**).



The cooling curves for these samples is below the heating curves, indicating oxidation of maghemite/magnetite, likely to hematite which has a lower spontaneous magnetization. The presence of pyrite is evident in a number of marls and limestones, where the increase in the magnetization at 390-420°C indicates that pyrite transforms to magnetite producing an intensity maximum at 480-500°C. Above 500°C, the newly formed magnetite is subsequently demagnetized or oxidized at

580°C (Fig. 4a, d, g). The cooling curves below 400°C are higher than the heating curves because of this newly formed magnetite which causes the spurious demagnetization behaviour at temperatures above 400°C. Pyrite is not always present in the marls, as is evident from marly samples that show a continuous decay until ~580°C, indicating (T-poor) magnetite only (Fig. 4c,f,i). Sometimes the amount of magnetite is very small and the Curie curve shows a mostly paramagnetic shape (Fig. 4c,i). Nevertheless, some magnetite must have been present since the cooling curve is lower than the heating curve because

of demagnetization/oxidation of the magnetite.

### 4.2 Demagnetization

Samples with magnetite-hosted magnetizations showed decreasing magnetization up to 550-580°C, or fields of 60-100mT. Hematite-bearing samples required higher temperatures up to 680°C. Generally, low temperature/low coercivity overprints were minor. Many samples and sites yielded erratic demagnetization behaviour yielding no interpretable paleomagnetic

results (e.g., **Fig. 5oo, pp**) and were discarded from further analysis. In successful sites, a minor viscous component was generally removed at low temperatures (100-150°C) or at low alternating fields (~10 mT). A low temperature/low coercivity component with present-day field direction is generally removed at 200-240°C or 20-30 mT (e.g. **Fig. 5q, r**). Where no linear decay of a component towards the origin was identified, we used great circle interpretations (e.g, **Fig 5g, h, hh).**

### 4.3 Paleomagnetic results

Below, we describe the paleomagnetic directions obtained from each of our 21 localities. Results are summarized in **Table 1** and locality averages are illustrated in **Figs. 2 and 3**. All demagnetization diagrams per site are provided in the online appendix as *site*.dir files that can be imported into Paleomanetism.org. Statistical analysis was first performed on a per site basis (**Table 1**), whereby outliers were omitted (but are listed in **Table 1** and provided Paleomagnetism.org-compatible files). Subsequently, individual directions from the 21 localities were grouped and averaged (**Figs. 2, 3, 6; Table 1**). Locality

averages were calculated using individual sample directions, such that large sites (large number of samples) have a larger statistical weight than smaller ones, following procedures in (Deenen et al., 2014, 2011). Locality results are shown as equal area projections of ChRM directions and their means before and after tectonic correction (**Figure 6, Table 1**). All locality results are provided in the online appendix and can be visualized and used in Paleomagnetism.org.



### 4.3.1 Ulukışla basin

Within the Ulukışla basin, we collected ~1200 samples from 49 sites, creating 13 localities consisting of Upper Cretaceous to Upper Miocene rocks (**Figs. 4, 6; Table 1**). Here, we describe these localities from stratigraphically old to young.

Around *Alihoca*, in the south of the basin (**Fig. 2**), we collected four sites in Upper Cretaceous red hemipelagic limestones (AL1, AR2) and from blue limestones (AL3, 4) (Alihoca-1 locality). In addition, we collected one site (AL2) from Upper Paleocene silty marls (Alihoca-2 locality). An overprint in the red limestones was removed at temperatures of ~200°C, demagnetization at higher temperatures up to ~500°C showed a linear decay towards the origin (**Fig. 5e, f**), at higher temperatures magnetization behaviour is erratic. AF-demagnetization follows great circles and a reliable ChRM direction

(setpoint) decaying linearly to the origin was not obtained (**Fig. 5g, h**). We interpret this as a magnetization carried by magnetite as well as hematite. The blue limestones yielded directions coinciding with the present-day field (PDF) in geographic coordinates (AL4), or erratic demagnetization behaviour due to very low intensities (AL3); both were discarded. Sites AL1 (reversed) and AR2 (normal) do not give a positive reversal test, and an optimal clustering of their directions at 130% unfolding (**Fig. 7a**) provides an indeterminate (but not negative) fold test. Both sites give A95 values that are just

above A95$_{min}$, showing that they may be explained by PSV, but only barely. The Paleocene site (AL2) yielded mainly great circle trajectories for AF fields up to 40mT (and displayed gyroremanence at higher coercivities (Dankers and Zijderveld, 1981), and for temperatures up to ~500°C; the few set points showed a strongly rotated reversed field. All sites from the Upper Cretaceous as well as from the Upper Paleocene rocks of the Alihoca locality consistently suggest large rotations as much as 90ccw (**Fig. 6**), despite the indeterminate field tests.

*The Ardıçlı* locality was sampled in two sites (AR1, AR3) along a road section along the canyon to Ardıçlı village in the southeast of the Ulukışla basin (**Fig. 2**). The lithology is a dark grey sandy-silty limestone of Campanian-Maastrichtian age. A PDF component was resolved at temperatures up to ~200°C and fields of ~50 mT. In most cases, linear decay towards the origin occurred at temperatures up to ~320°C suggesting an iron sulphide as carrier of the magnetization. AF-

demagnetization does not yield full demagnetization at fields of 100 mT. The thermal and AF behaviour is consistent with the presence of fine-grained pyrrhotite which has very high coercivities (Dekkers, 1988). A total of 11 (out of 34) samples gave noisy demagnetization data from which no direction was isolated. We interpret the component isolated between 200 and 350°C as the ChRM; all being reversed. The two sites share the same bedding and a fold-test is thus not possible. The direction in tectonic coordinates suggests a vertical axis rotation of 45.9±2.6°cw. We note, however, that A95 (2.5) is lower

than the A95$_{min}$ (3.4), indicating under-sampling of PSV. In addition, the tilt-corrected inclination of ~30° is considerably lower than that for Eurasia in the Late Cretaceous-Paleogene (~50°), whereas the geographic inclination of ~45° is not. Together, this indicates that the Ardıçlı locality is remagnetized after tilting and has undergone a significant rotation after remagnetization: directions in geographic coordinates yields a rotation of ~80°cw (**Fig. 6**, **Table 1**).





The *Bekçili* locality consists of six sites collected from uppermost Cretaceous to Upper Paleocene marls and interbedded turbiditic sandstones (Camardi Formation) in the northern part of the Ulukışla basin (**Fig. 2**), between the villages of Bekçili and Üskül. A PDF is generally removed at 15 mT or ~200-250°C. Subsequently, a component linearly decaying to the origin

between 20 and 50 mT or 300 and 560°C is interpreted as the ChRM (**Figure 3K,L**) likely carried by magnetite. The A95 of the mean lies within the A95$_{min-max}$ envelope, the Bekçili locality therefore provides a well defined rotation of 35.4±2.9°ccw (**Fig. 6**, **Table 1**).

The *Kızılkapı* locality is based on five sites sampled in Upper Paleocene marls and siltstones in the north of the basin

between Kızılkapı and Postallı villages (**Fig. 2**). A PDF is generally removed at 20 mT or ~220-240°C. A component linearly decaying to the origin between 20 and 50 mT or 275 and 450°C is interpreted as the ChRM, likely carried by magnetite. At higher temperatures the magnetization becomes erratic because of the oxidation of pyrite. Site PC3 provided only erratic demagnetization behaviour and was discarded. Site PC1 yielded an inconsistent direction in both geographic and tectonic coordinates (D/I = ~52/-47° and 357/-65°, respectively), and was discarded from further analysis. The remaining

sites CP1, PC2, and PC4 yield a positive fold test suggesting a primary magnetization, but yields inclinations that are somewhat steeper than predicted for the Eurasian reference (~64±4° vs. ~50°). The mean direction suggests a rotation of 13.2±7.2° cw (**Fig. 6**, **Table 1**).

The *Kolsuz* locality consists of two sites collected from Paleocene marine marls interbedded with volcanic rocks in the west

of the basin (**Fig. 2**). A PDF is generally removed at 20 mT and ~180°C and linear decay towards the origin or great circle trends occur between 20 and 50 mT and 200 and ~450°C, suggesting magnetite as carrier. The two sites both yield reversed polarity directions with shallow inclinations of ~30°, both before and after tilt correction, which may be explained by compaction-induced inclination shallowing. A fold test is positive (**Fig. 7c**). The two sites combined yield an A95 value within the A95$_{min-max}$ envelope. The locality suggests a rotation of 24.2±7.8° ccw (**Fig. 6**, **Table 1**).

The *Halkapınar* locality consists of nine sites collected from Paleocene to middle Eocene siltstones and marls in the southwest of the basin (**Fig. 2**). Site HP5 provided erratic demagnetization behaviour, while site HP4 yielded an unrealistically shallow inclination of 9°; both sites were discarded. Site KG1 provided a direction that in geographic coordinates coincides with the PDF, and in tectonic coordinates yielded a very low inclination of ~15°; we interpret the site

as a recent overprint and discard it. Site YL1 provides a tightly clustered set of reversed directions decaying towards the origin at low temperatures of up to 200°C, with an A95 (4.3°) lower than A95$_{min}$ (6.3°). The inclination in geographic coordinates (-53±6°) coincides with the expected inclination for the sample locality, whereas in tectonic coordinates it is too low (-25±8°). Site HP4 yields a very low inclination in tectonic coordinates of ~9° and a very tight clustering (K=2180) with A95 << A95$_{min}$ suggesting remagnetization. We interpret sites YL1 and HP4 hence as remagnetized, and discard them from





further analysis (**Table 1**). Of the remaining five sites, KG2 and HP2 have normal polarity, and HP1, HP3 and HP6 have reversed polarity. A PDF direction was eliminated at temperatures of ~100-150°C and fields of 10 mT. The ChRM components were interpreted between 150-600°C and fields of 15-80 mT. The primary carrier of the magnetic signal is magnetite. The fold test shows optimal clustering at ~61-95% unfolding (**Fig. 7d**). The Halkapınar sites come from a

sedimentary sequence that shows evidence for syn-folding sedimentation: the lower part of this sequence is tilted steeper to the north than the upper part (Gürer et al., 2016), consistent with the fold-test, and we hence interpret the magnetization as primary. The A95 value is within the $A95_{min-max}$ envelope (Deenen et al., 2011) suggesting representation of PSV. The locality thus provides a well-defined rotation, of 26.0±5.1°ccw (**Fig. 6**, **Table 1**).

The *Eminlik* locality was collected at four sites in Middle Eocene sandy and silty marls exposed in the Ovacık syncline (**Fig. 2**). A minor viscous component was removed at temperatures up to ~150°C. The ChRM of sites was interpreted largely between ~200-450°C and ~10-30mT. EM1 and EM5 show a declination and inclination indistinguishable from the present day magnetic field (PDF) and were discarded from further analysis. Site EM2 yielded both normal and reversed, antipodal directions that do not decay towards the origin, with steep inclinations close to today's (~56°) and declinations indistinguishable from north and south, respectively, but with unrealistically shallow inclinations in tectonic coordinates

(~8°). Similarly, site EM4 contains reversed components that do not decay towards the origin with a steep inclination before, but a very shallow inclination after tilt correction. We interpret these directions to reflect a post-tilt remagnetization of reversed and normal polarity and discard the sites from further consideration. The four sites of the Eminlik locality hence did not provide a rotation estimate (**Fig. 6**, **Table 1**).

The *Hasangazi* locality was collected from on six sites sampled in grey sandy limestones and silty marls of Eocene age between Teknecukur and Hasangazi villages (**Fig. 2**). A minor viscous component was eliminated at temperatures up to 180°C. Three sites from this locality (HG1, GM2, and TC1) are indistinguishable from the present day magnetic field (PDF) in geographic coordinates and were discarded from further analysis (**Table 1**) and site GM1 yielded erratic demagnetization

behaviour. For the remaining sites the component from 280-350°C; 20-60mT (TC2, **Fig. 4s,t**), and 320°C-450°C; 20-60mT (HG2) was interpreted as the ChRM, or used to construct great circles. Sites HG2 and TC2 defining the Hasangazi locality (n=33) yielded a positive fold test (**Fig. 7e.** The A95 value lies just below the $A95_{minx}$ value (**Table 1**) which likely results from the use of great circles that seek optimal clustering. The fold test shows optimal clustering at 84-96% which we consider as positive (**Fig. 7e**). We obtain and a declination indicative of 4.9°±2.8°cw rotation (**Fig. 6**, **Table 1**).

The *Topraktepe* locality was collected at a single site in Middle Eocene silty, dark limestones and marls in the central part of the basin (**Fig. 2**). A component up to ~180°C and 20 mT carries the PDF. Linear decay towards the origin follows until ~360°C interpreted as a magnetization carried by iron sulfides, and strong, erratic magnetizations appearing at 390-420°C is interpreted as the breakdown of pyrite into magnetite (**Fig. 4d**). The ChRM was interpreted from the linearly decaying





component between ~200-350°C and ~25-40mT. The resulting scatter of the 28 directions isolated from Topraktepe yields an A95 value within the A95$_{min-max}$ envelope. The inclination in geographic coordinates is lower (~27°) than in tectonic coordinates (~45°). We interpret the direction as primary, and the locality suggests a rotation of 44.5±4.1° ccw (**Fig. 6**, **Table 1**).

The *Aktoprak* locality contains four sites sampled in Chattian (Upper Oligocene) continental brown and red silt and sandstones exposed in the Aktoprak syncline in the southwest of the basin. Curie balance measurements (**Fig. 4b)** identify magnetite as the main magnetic carrier. Site AT1 yielded erratic demagnetization behaviour and site AT4 yielded an average direction indistinguishable from PDF in geographic coordinates (**Table 1**). Site AT2 (n=16) did not yield components

linearly decaying towards the origin. Great circles interpreted from the demagnetization diagrams do not have a common intersection line, and no direction was interpreted from this site. Site AT3, finally, provided a component linearly decaying towards the origin, or defining great circle trajectories, between 200 and 570°C and 20-60 mT interpreted as the ChRM. This site suggests a rotation of 24.5±10.1°. Meijers et al. (2016) reported paleomagnetic data from the same stratigraphy based on a much larger dataset of >120 directions that is similar to our result from AT3. Combining these results, whereby we

parametrically sampled the Meijers et al. (2016) dataset, yields a dataset with an A95 value within the A95$_{min-max}$ envelope, with a mean direction suggesting a rotation of 33.7±3.8°ccw (**Fig. 6**, **Table 1**).

The *Postallı* locality is based on two sites (CP2 and CP3) sampled in red continental sandstones and siltstones, and a light-coloured tuffaceous sandstone, respectively. Both sites are of (Upper?) Miocene age and are located in the northern part of

the basin, north of Postallı village and dam (**Fig. 2**). In both sites a PDF overprint is removed at temperatures of up to ~180°C. The main magnetic carrier of samples in site CP2 is magnetite since the component decaying to the origin and interpreted as ChRM lies between 180°C and 500-580°C. AF-demagnetization shows a small or virtually no decrease of the NRM, which suggest a high-coercive magnetic mineral. Hematite is ruled out because of the thermal demagnetization characteristics. It could be goethite but also this is not readily evident from thermal treatment, so possibly some pyrrhotite is

present that may give high coercive forces (Dekkers, 1988), like in locality Ardıçlı. AF-demagnetization did not lead to full demagnetization, but components clustering or decaying towards the origin between 25 and 60 mT were interpreted as the ChRM (**Fig. 5u**), and are consistent with those form the thermal treatment. Site CP2 contains reversed directions. Before tectonic correction the inclination is steep (68.6±9.0°), and becomes shallower (43.3±12.0°) upon tectonic correction. Site CP3 hosts a component that was interpreted as ChRM between ~210-430°C and ~25-60 mT likely carried by magnetite. This

site contains both normal and reversed polarity, but with insufficient samples for a meaningful reversal test, while also the fold test is indeterminate (maximal clustering between ~50 and 150% unfolding). The two sites combined suggest a vertical axis rotation of 17.7 ± 5.8°ccw (**Fig. 6**, **Table 1**).



The *Burç* locality contains one site sampled in Upper(?) Miocene lacustrine sandstones and silts in the northeast of the basin, south of Çamardı (**Fig. 2**). This site only provided erratic demagnetization behaviour, from which no consistent ChRM component was interpreted. Hence, this site was discarded from further analysis (**Table 1**).

The *Hacıbekirli* locality contains one site sampled in Upper Miocene terrestrial sandstones and silts in the southwestern part of the basin (**Fig. 2**). A PDF overprint is removed at temperatures of ~150°C. Components decaying towards the origin between 10 and 30 mT, and 150-350°C were interpreted as the ChRM. The site yields a reversed magnetization with a tight clustering of interpreted ChRM directions (A95 = 3.0, A95$_{min}$=3.8). This may indicate remagnetization. In geographic coordinates, a direction of ~182/-53° yields an inclination close to the expected inclination for the region, whereas in tectonic

coordinates a direction of 165/34° yields a much lower inclination suggesting major inclination shallowing (**Table 1**). In absence of positive field tests, we refrain from interpreting this direction as primary even though the rotation suggested in tectonic coordinates is similar to that from other Miocene localities (**Fig. 6**, **Table 1**).

Finally, we calculated an average paleomagnetic direction for the Upper Miocene to Pliocene volcanic rocks of *Cappadocia*

by combining lava sites previously published by (Çinku et al., 2016; J D A Piper et al., 2002; Piper et al., 2013; Platzman et al., 1998). These yield a mean (n=77) with an A95 (2.8) within the A95$_{min-max}$ envelope (2.1, 5.3) suggesting that PSV is sufficiently represented. These data suggest a vertical axis rotation of 6.5±3.3°ccw (**Fig. 1; Table 1**).

### 4.3.2 Tauride fold-and-thrust belt and overlying basins

115 samples from three localities (Berendi, Aladağ, Gürün) consisting of 11 sites were sampled in the Eocene to Miocene

stratigraphy of the sediments overlying the Taurides fold-and-thrust belt (**Figs. 2, 3**; **Table 1**). In addition, one locality (Sariz) with 4 sites and 63 samples was sampled from folded and thrusted Late Cretaceous-Eocene limestones within the eastern Tauride fold-thrust belt. The maps (**Fig. 2,3**) show the average declination of each locality with its associated ΔD$_x$, whereby all directions were calculated into normal polarity.

The *Berendi* locality is based on six sites sampled from shallow marine nodular limestones on the Taurides, southwest of the Ulukışla basin (**Fig. 2**). We obtained calcareous nannofossils from this locality (see supplementary information), showing a Thanetian (Late Paleocene) to Ypresian (Early Eocene) age. Site BR5 gives a direction indistinguishable from a PDF. Site BR6 yielded erratic demagnetization behaviour. From site BR1, a component linearly decaying towards the origin between 200-450°C, and 15-45mT was interpreted as the ChRM, or defined by great circles. A fold test on sites BR1-4 is clearly

negative (**Fig 7f**), implying a post-folding magnetization. The remagnetized direction in geographic coordinates has a declination of 343.1±6.2°, suggesting a post-folding rotation of ~17°± 6° ccw (**Fig. 6**, **Table 1**).





The *Aladağ* locality is based on four sites sampled in folded Oligocene continental red sandstones and silts of the Karsanti basin overlying the Taurides (Ünlügenç et al., 2015) near Aladağ village east of the EFZ (**Fig. 2**). The Lower Miocene of the Adana basin unconformably overlying the Karsanti basin is not folded. The magnetisation of these sites is primarily carried by magnetite, with the ChRM generally interpreted between 10-40mT. Sites AD1 and AD3 did not result in
paleomagnetically meaningful directions. Sites AD4 and AD5 yield a negative fold test (**Fig. 7**) showing that they carry a post-tilt magnetization, acquired sometime after the Oligocene. This is further suggested by the shallow inclinations upon tectonic correction (~30° compared to ~50° in geographic coordinates). The combination of these two sites yields a declination in geographic coordinates of 343.4±7.3°, suggesting a ccw rotation of 16.6±7.3 (**Table 1**), with an ill-defined but post-folding age.

The *Sarız* locality consists of four sites, one in Upper Cretaceous and three in Eocene limestones sampled in the eastern Tauride fold-thrust belt to the northeast of Sarız (**Fig. 3**). In all sites, a PDF component is removed at temperatures up to 180°C, or up to ~25mT, followed by erratic demagnetization behaviour. The Sariz locality yielded therefore no reliable directions and must be discarded.

The *Gürün* locality was sampled at one site in Miocene mudstones overlying the Eastern Taurides (**Fig. 3**). A PDF component was removed at temperatures of 180°C. AF-demagnetization did not yield any paleomagnetically meaningful interpretations. In tectonic coordinates all interpreted samples show a reverse polarity, which results in an average declination of 163.9±10.9° after tectonic correction, constraining a rotation of 16.1°±10.9° ccw (**Fig. 6**, **Table 1**). No field
tests can be applied to this site (SS10) and we cannot conclude that this site carries a primary magnetization. In geographic coordinates, however, the declination is 146.9±12° (**Fig. 6**, **Table 1**), suggesting a larger ccw rotation of ~33°.

### 4.3.3 Sivas basin

Within the Sivas basin, we collected ~900 samples from in total 51 sites defining 12 localities consisting of Paleocene to Upper Miocene rocks of the Sivas basin (**Fig. 3; Table 1**). Additionally, several published magnetostratigraphic sites from
the western part of the basin were re-interpreted (Krijgsman et al., 1996; Langereis et al., 1990). One locality of three sites was sampled on the Kırşehir Block, northwest of the basin. Here we describe the localities from stratigraphically old to young, their demagnetization behaviour and the resulting rotations of successful localities.

The *Ulaş* locality comprises nine sites from Paleocene-Eocene marls around Ulaş in the central west part of the region, ~30
km south of Sivas city (**Fig. 3**). Site SS11 yielded a low-temperature PDF component followed by erratic demagnetization behaviour. Sites SS18, SS21 and SS31 yielded great circles with no clear intersection and were discarded. Site SS20 yielded reversed directions and great circles that provide a very low, downward inclination in tectonic coordinates (13°) and was



discarded from further analysis. Site SS31 provided a well-defined component demagnetizing towards the origin that yielded a direction coinciding with PDF in geographic coordinates, but with an anomalously steep inclination (~75°) in tectonic coordinates and was discarded from further analysis. The remaining sites have reversed directions with in most cases a PDF that was removed by 15mT or 180°C, although in some samples the entire magnetization up to 60 mT or 580°C showed a

direction coinciding with a PDF in geographic coordinates. ChRM components were interpreted between 10-70 mT and 250-580°C. The three remaining sites yielded a negative fold test suggesting remagnetization in the late stages of folding (or between two folding phases). In geographic coordinates, the distributions of ChRM and VGP are circular, but in tectonic coordinates very elongated, in line with the negative fold test. The declination in geographic coordinates (8±7°) (**Fig. 6**, **Table 1**) shows a hardly significant cw rotation since remagnetization.

The *Akkışla* locality comprises eight sites of Eocene age in the southwest of the basin, unconformably overlying Afyon zone metamorphic rocks east of Akkışla (**Fig. 3**). All sites were sampled from marls. Two of the sites (SS1 and SS25) have a reversed and six have a normal polarity. Sites SS4 and SS56 gave a direction indistinguishable from PDF in geographic coordinates and were discarded from further analysis. In the remaining 6 sites a PDF overprint is removed at temperatures of

up to ~180°C. Components decaying to the origin were interpreted as the ChRM between a range of ~12-60 mT and 180-500°C, and are likely hosted by magnetite. The six remaining sites yield positive fold test **(Fig. 7i)** and yield a declination suggesting a rotation of 27.6±3.7°ccw (**Fig. 6**, **Table 1**).

The *Gürlevik* locality was sampled at five sites from Eocene marls north of Gürlevik mountain, between Sincan and Zara in

the central south of the basin (**Fig. 3**). The magnetisation of these sites is primarily carried by magnetite, with the ChRM interpreted between generally between 150-370°C and 20-65mT. Sites SS36 and SS45 yielded erratic demagnetization behaviour and site SS43 only provided a low-temperature/low-coercivity PDF direction. Sites SS33 and SS44 give a positive fold test **(Fig. 7j)** and have an A95 value within the A95$_{min-max}$ envelope. The declination shows a rotation of 19.8±9.5°cw (**Fig. 6**, **Table 1**).

The *Güllük* locality was sampled at five sites from Upper Eocene to Oligocene red sandstones and siltstones in the central part and north of the study area, and on the northern flank of Tecer mountain (**Fig. 3**). The magnetisation of these sites is primarily carried by magnetite, with the ChRM interpreted between generally between 100-300°C and 10-60 mT.The fives sites provided generally high magnetic intensities with either components linearly decaying towards the origin, or, in most

cases, well-defined great circles. Those great circles define well-clustered intersections from which sample directions were estimated. Each of the five sites yielded a well-defined average paleomagnetic direction, but these five directions are strongly scattered. This may either reflect strong local rotations – which is unlikely given the absence of mini-basins or major faults within this locality. More likely, some of these sites may have been remagnetized by lightning, which would be consistent with their high intensities. Sites SS32 and SS52 give anomalously low inclinations <15° and site SS53 gives



declination suggesting a rotation of almost 100° derived from great-circle analyses only. Sites SS51 and SS54, on the other hand give a positive fold test (Figure 7). These two sites suggest a rotation of 37.6±10.7° ccw (**Fig. 6**, **Table 1**).

The *Akdağmadeni* locality consists of three sites sampled in Eocene marls and red continental deposits overlying the
easternmost part of the Kırşehir Block, northwest of the Sivas basin (**Fig. 3**). A PDF overprint is generally removed at fields of 15 mT and temperatures of 150°C. The primary carrier of the magnetization is magnetite. The sites are characterized by low intensities and unstable demagnetization behavior. Site SS14 provides mostly erratic directions, only a few samples with reversed polarity yielded an interpretable result, where the ChRM was interpreted between 20-45 mT, and 240-370°C. The interpreted directions and great circles suggest an unrealistically low inclination upon tectonic correction (-27°), compared to
in geographic coordinates (~54°). We hence interpret this site to represent a post-tilt magnetization which yields a rotation of ~20° ccw in geographic coordinates. Similarly, site SS15 provided some reversed directions, but mostly great circles that yielded no consistent intersection and the site was discarded from further analysis. A fold test of sites SS14 and SS15 was not possible. We note that the inclination of the sites in geographic coordinates is close to the one expected based on the European APWP, whereas in tectonic coordinates, the inclination is very shallow (22-35°), which indicates a post-tilt
magnetization. The two sites do not share a Common True Mean Direction. When combined, the sites yield a declination suggesting ~10°ccw rotation, but given the uncertainties above, we are not confident that this is geologically meaningful (**Fig. 6**, **Table 1**). Site SS16 only gave PDF directions and was discarded.

The *Sincan* locality is based on one site collected in the corridor of red Oligocene siltstones and sandstones between north of
Kangal and Divrigi towns. The primary carrier in this site (SS34) is magnetite. A PDF component was eliminated at temperatures up to 180°C and fields of 10 mT. The ChRM was interpreted between temperatures of 200-500°C and fields of 20-50 mT. The directions obtained from well-defined components decaying to the origin suggest a declination of ~280°, with a shallow inclination of ~27° in tectonic coordinates, and a declination of ~305°, with an inclination of ~49° in geographic coordinates. The shallow inclination in tectonic coordinates may indicate that this locality acquired its
magnetization after folding. The locality therefore suggests a major post-folding rotation of 55.7±9.0°ccw (**Fig. 6**, **Table 1**).

The *Yeniköy* locality consists of six sites sampled in Oligocene continental redbeds (fluvial and lacustrine siltstones and sandstones) in the southwestern part of the greater Sivas basin, approximately 20 km south of Sarkisla (**Fig. 3**). In addition, samples from 102 levels were collected from a 800 m long section of redbeds of the Yeniköy locality by (Krijgsman et al.,
1996). We have reinterpreted their data for rotation analysis. Almost all sites of this locality gave a direction in geographic coordinates which is indistinguishable from PDF. The remaining one site (SS58) that showed no PDF, and the ChRM is carried by hematite considering the temperature needed to fully remove the NRM, in accordance with the lithology (redbeds). The combined Yeniköy section and SS58 contain both normal and reversed directions. These yielded a positive



reversal test (Tauxe, 2010) and a positive fold test (**Fig. 7l**). The locality provides a rotation of 42.4±3.2°ccw (**Fig. 6**, **Table 1**).

The *Inkonak* locality consists of an Upper Oligocene series of fluvial and lacustrine sediments sampled by Krijgsman et al.

(1996) for magnetostratigraphic purposes. It was sampled ~50 km south of Sivas city (**Fig. 3**). The section is 313 m thick and consists of a regular alternation of clays, limestones and sandy deposits. We reinterpreted their demagnetization diagrams for tectonic purposes. A PDF component was removed at ~100°C, the ChRM was eliminated between 240-700°C. The section yielded both normal and reversed polarity directions, whereby approximately half of the reversed directions were interpreted using great circles. The reversal test was not positive in tectonic coordinates owing to an 8° difference in inclination,

whereas the declinations of both groups are identical in tectonic coordinates. In geographic coordinates, the reversal test is positive. There was insufficient variation in the bedding for a meaningful fold test. When combining the normal and reversed directions, a mean ChRM was obtained (n=87, A95=4.4, within the $A95_{min, max}$ envelope [2.0, 4.9]) and shows, in tectonic coordinates, that the inclination becomes steeper and closer to the expected value at the site location (~34 and ~46° in geographic and tectonic coordinates, respectively, see Table 1). We tentatively interpret the magnetization as primary, which

suggests a large rotation of 56.0±5.0°ccw (**Fig. 6**, **Table 1**). If the magnetization is secondary, the locality would give a post-remagnetization rotation of 45.7±4.2°ccw, which may thus be regarded as a minimum value.

The *Gemerek* locality was sampled in the western part of the basin (**Fig. 3**) and analysed by Krijgsman et al., (1996) for magnetostratigraphic purposes and was obtained from a continuous stratigraphic section of 200 m consisting of middle

Miocene (~15 Ma) fluvial/lacustrine sediments and a basalt. We have re-analyzed the data. The samples recovered from the sediments yield a PDF component, which was removed at 100-240°C. The ChRM component was interpreted at temperatures up to 700°C. The thermal demagnetization of the basalt revealed a normal component at temperatures between 200-300°C and 450-580°C, and a reversed component at temperatures between 300-450°C (Krijgsman et al., 1996). Together, the Gemerek locality contains normal and reversed intervals generally yielding well-defined components decaying

towards the origin. The normal and reversed directions share a positive reversal test in tectonic coordinates (Tauxe et al., 2010) and the directions (without great circle interpretations) yield a positive fold test (**Fig. 7m**). We therefore consider the magnetization as primary and interpret a rotation of 49.9±3.8°ccw for this locality (**Fig. 6**, **Table 1**).

The *Bünyan* locality consists of one site collected from white lacustrine marls of Miocene age in the west of the basin (**Fig.**

**3**). The samples of this site (SS24) showed a strong PDF overprint, with some of them providing great circles, which did not provide a conclusive intersection. The locality was discarded from further analysis (**Table 1**).

The *Sivas* locality was sampled along road cuts on both sides of the main road to Sivas city, between Kovali and Güllüce (**Fig. 3**). All four sites were sampled from Miocene continental redbeds made of sandstones and silts. Site SS12 yields five





reversed directions based on high-temperature components decaying towards the origin. In addition, the site yields 20 parallel great circles, whereby the interpreted ChRM directions lie on the great circle trajectories. Interpreting directions on the great circles coinciding best with the five interpreted directions would lead to a very highly clustered, n=25 average that we feel is not substantiated by the quality of the demagnetization diagrams, and we only use the five directly obtained ChRM

directions for our average. SS13 yielded well-defined great circles, yielding a well-defined intersection that we interpret as the ChRM. Site SS22 showed a PDF overprint removed at temperatures of around 180° and fields of 5mT. A normal component interpreted as ChRM was found between 300-600°C, and fields of 8-40mT. The resulting direction yielded in geographic coordinates an inclination close to the expected one for the locality, but a very shallow one in tectonic coordinates, which may indicate post-tilt remagnetization. If so, the site rotated ~20°cw after remagnetization, prior to

acquiring a PDF. Site SS23 yielded two interpreted ChRM directions and a set of great circles with well-defined intersection providing the site average. The four sites yield a positive fold test, but with large uncertainties. Moreover, this fold test is strongly influenced by the five directions of site SS12. These show a declination strongly deviating from the other three sites. Performing the fold test only on SS13, SS22 and SS23 yields a negative fold test (**Fig. 7n**) suggesting post-folding remagnetization. Moreover, the inclinations (53-59°) of these three sites are all coinciding with the expected one in

geographic coordinates, but are significantly shallower in tectonic coordinates (27-44°). We thus suspect a remagnetization after (most of) the folding and interpret a post-remagnetization rotation of 16.3±4.8° cw (**Fig. 6**, **Table 1**).

The *Zara* locality consist of three sites sampled along road sections around Bulucan ~20 km south of Zara town (**Fig. 3**). Sites SS35 and SS42 were sampled in Miocene siltstones and sandstones. Site SS37 was collected from shallow marine

marls and sandstones. The latter yielded a low-temperature, low-coercivity component coinciding with PDF in geographic coordinates, followed by erratic demagnetization behaviour. Site SS35 yielded in geographic coordinates a reversed direction with an inclination coinciding with the expected inclination for Eurasia (185/-58), with an A95 much lower than $A95_{min}$ (2.8 vs. 4.6°), and has likely been remagnetized. In tectonic coordinates the site yielded a paleomagnetically meaningless direction of D/I = 355/-82°. We interpret this site as remagnetized after folding, after which no significant rotation occurred.

Site SS42 yielded only great circles with no clear intersection and was discarded (**Table 1**).

The *Kemah* locality consists of four Miocene sites sampled in sandstones and siltstones north of Kemah in the far east of the study region. Magnetite was identified as the primary magnetic carrier. A PDF component was removed between 100-150°C and fields of 20 mT. Site SS38 was mostly interpreted with great circle analysis, which yielded no clear intersection. Site

SS40 yielded strong PDF overprints and great circle trajectories. One isolated reversed direction did not fall on the consistent great-circle trajectories and the site was discarded. SS39 yielded strong PDF overprints, but these did not converge towards the origin and defined great circles with an optimal intersection in geographic coordinates around 188/58°, i.e. with an inclination close to the expected inclination at this location. Site SS41 also yields only great circles with a well-defined




intersection. Sites SS39 and 41 yield a negative fold test (**Fig. 7**). We interpret the combined direction of SS39 and SS41 as a post-folding magnetization that shows a post-remagnetization rotation of 6.7±4.3° cw (**Fig. 6; Table 1**).

The *Kalkar* locality consist of the Kaleköy and Karaözü sections, and was sampled for magnetostratigraphic purposes by
Langereis et al. (1990) in Upper Miocene (latest Vallesian; Sümengen et al., 1990) continental sediments. The continental sediments consist mainly of silts and brown clays with occasionally thick sand layers. We reinterpreted the demagnetization diagrams for tectonic purposes. A large overprint is only removed at relatively high temperatures (above 300°C) and a final component towards the origin is only removed at high temperatures (610-650°C), pointing to maghemite and/or hematite as the main carrier of the ChRM. The Kalkar section yielded normal and reversed polarity directions that share a positive
reversal test in geographic coordinates but a negative one in tectonic coordinates. Although in tectonic coordinates the mean declination (~336±6°) shows a 24±6°ccw rotation, the tilt corrected inclination is much shallower (~34°) than in geographic coordinates (60°). Likely, the locality has been remagnetized after folding and shows no significant rotation (Table 1).

## 5 Discussion

### 5.1 Regional versus local rotation in the Ulukışla basin

The sampling of the Ulukışla basin aimed to evaluate whether there is a significant rotation difference between the basin and the southern part of Kırşehir block. Our results show that there are variations in the declinations derived from localities across the Upper Cretaceous to Upper Miocene stratigraphy of the Ulukışla basin (**Fig. 2, Table 1**). In the following we will discuss the meaning of each successful locality and evaluate the regional rotation of the basin, and discuss differences in the context of the major structures within the basin.

The majority of localities, covering Paleocene to Oligocene sediments, contain primary magnetizations that display evidence for counter-clockwise rotations on the order of ~20-40°. These include from E to W the Bekçili, Topraktepe, Kolsuz, Aktoprak and Halkapinar localities. Together these localities provide an Oligocene or younger, counter-clockwise rotation of the basin of 32.3±2.2° (n=326, K=16.4) when all individual directions are averaged, or 32.2±9.0° (n=5, K=84.5) when locality results are averaged.

In the southeastern parts of the basin, localities show rotations that strongly deviate from this average. The Upper Cretaceous and Paleocene sediments of the Alihoca locality in the southeast of the basin, overlying ophiolite yielded a very large counter-clockwise rotation of ~90°. Gürer et al., (2016) showed that the southern sediments of the basin revealed phases of rapid uplift and subsequent subsidence in the Late Cretaceous. The uplift-subsidence-uplift-subsidence cycle was interpreted as the response to the underthrusting of the continental Kırşehir block causing forearc uplift, then the potentially oceanic
intra-Tauride basin causing forearc subsidence, and finally the Afyon zone margin of the Tauride block causing renewed uplift (Gürer et al., 2016; 2017). Paleocene subsidence was then accommodated along a major normal fault that bounds the basin to the south and that exhumed the Afyon zone (Gürer et al., 2017), exposed immediately south of the Alihoca



localities. Finally, the Alihoca locality is located close to the strike-slip EFZ. The strong local rotations in the Alihoca ophiolite are likely caused by these strong tectonic motions and structures, and are not representative for the Ulukışla basin. The Ardıçlı locality was sampled in Upper Cretaceous sediments in the immediate hanging wall of a thrust where also a series of smaller strike-slip faults are found (Gürer et al., 2016) (**Fig. 2**). This locality yielded a post-folding clockwise

rotation of ~80°. The very low dispersion (k=147, A95<A95$_{min}$) suggests under-sampling of PSV and may be caused be later intrusion of mafic dykes into the Cretaceous stratigraphy of the locality. We consider also the Ardıçlı rotation result as not regionally representative. Given the strong deformation, remagnetization (Ardıçlı) and indeterminate fold test (Alihoca), we interpret these localities to record strong local rotations. The Upper Paleocene Kizilkapi sediments yielded counter-clockwise rotations of ~45° before and clockwise rotation of ~13° after tilt correction. As discussed above, inclinations are to

steep after tectonic correction while the post-folding counter-clockwise rotation agrees well with the regional pattern. Finally, the Hasangazi locality of Eocene age defines a minor clockwise rotation. The sampled sites lie in a tightly folded plunging syncline in the footwall of the central top south-verging thrust **(Fig. 2;** Gürer et al., 2016). Hence, we interpret this rotation, even though well-defined, as a local rotation associated with the strong local deformation.

Finally, the available paleomagnetic information from Miocene stratigraphy show a smaller rotation than the Paleocene to

Oligocene localities. Our compilation of previously published paleomagnetic results from the Upper Miocene to Pliocene rocks from the Cappadocia volcanic region, sampled across a large area of the southern Kırşehir block (Çinku et al., 2016; J D A Piper et al., 2002; Piper et al., 2013; Platzman et al., 1998) yielded a minor counter-clockwise rotation of 6±3°, suggesting significant rotations predated the Late Miocene. The poorly dated but presumed Miocene Postalli locality in the northern part of the basin suggests a counter-clockwise rotation of 17.7±5.4°, suggesting that part of the rotation may have

extended into the Miocene, although the small areal coverage of the site cannot exclude a local rotation origin.

Lefebvre et al. (2013) obtained paleomagnetic data from 4 sites in Cretaceous granitoids in the southern Kırşehir block (their Ağacören-Avanos Block, or AAB) and concluded a 28-35° counter-clockwise rotation. We have recalculated the average direction of these four sites by averaging all individual directions, which yields a counter-clockwise rotation of 28.7±2.8° (n=248, K=14.8) (**Table 1**). Comparing this number to our average from Ulukışla yields a negligible rotation difference of

3.6±3.6°, suggesting that the Ulukışla basin and southern Kırşehir block (the AAB of Lefebvre et al. (2013)) form one coherently rotating domain, whereby our Ulukışla data suggest an age of rotation sometime in or after the Oligocene, younger than the Paleocene-Eocene age postulated by Lefebvre et al. (2013). This age is consistent with the age of the Savcılı Thrust Zone (STZ; Lefebvre et al., 2013; Advokaat et al., 2014; Isik et al., 2014), which bounds the southern Kırşehir- Ulukışla rotating domain to the north. This thrust zone may continue into the Kursunludag Thrust Zone in the east

(Dirik and Göncüoglu, 1996).

**5.2 Relationship of rotation in the Central and Eastern Taurides with the Ulukışla basin and the southern Kırşehir Block**



No paleomagnetic data are available from the carbonates of the Taurides in the Bolkardag mountains immediately to the south of the Ulukışla basin (and to the east of the Berendi locality) that could test whether the Central Taurides are part of the same rotating domain. Our results from the Eocene sediments of the Berendi locality overlying the Central Taurides shows that 17±6° ccw rotation occurred following a post-folding remagnetization. We cannot constrain the timing of the

folding or the remagnetization, which may thus well have occurred anytime since the (post-)Oligocene ~30°ccw rotation of the Ulukışla basin and the AAB. The Berendi results, however, do suggest that the Bolkardag mountains were not part of the clockwise rotating domain documented in the west-central Taurides to the northwest of the Mut basin (Çinku et al., 2016; Kissel et al., 1993; Meijers et al., 2011). The Bolkardag mountains are separated by a fault from the Ulukışla basin that was previously interpreted as a major back-thrust (Blumenthal, 1956; Demirtasli et al., 1984). Gürer et al. (2016; 2017), however,

showed that this fault is a folded and overturned normal fault that was sealed by Eocene sediments. Although this normal fault may have accommodated a rotation difference between the Ulukışla basin and the Bolkar mountains during its Paleocene to Early Eocene activity, it is unlikely that it accommodated major differential rotation differences since the middle Eocene. The formation of the Bolkardag? fold in Oligocene time was associated with no more than a few kilometres of shortening, which is unlikely to have resulted in a major differential rotation. We therefore conclude that the Bolkar

mountains formed a coherent part of the counter-clockwise rotating domain together with the Ulukışla basin and the southern Kırşehir Block.

Paleomagnetic data of Cinku et al. (2016) show that immediately east of the EFZ, the amount of rotation measured in Upper Cretaceous carbonates increases slightly towards the fault. Combining all their (parametrically sampled) sites yields a declination of 318.5±3.0°, suggesting a net counter-clockwise vertical axis rotation of 41.5±3° (n=154, K=16.1) **(Fig. 1,**

**arrow with reference 11)**. Farther towards the east, where the strike of the eastern Taurides changes from NE-SW to ENE-WSW, the paleomagnetic data from the Tauride units are sparse, but four sites of Kissel et al. (2003) and Cinku et al. (2016) (**Fig. 1,** arrows with references 4 and 11, Eastern Taurides locality in **Fig. 3**) from Eocene limestones yield a declination of 329.8±3.8° (n=80, K=22.3), suggesting a 30.2±3.8° ccw rotation. Our Eocene Akkisla locality immediately north of the eastern Taurides (**Fig. 3**) showed a 27.6±3.7° ccw rotation, within error identical to that of the Eastern Taurides carbonates.

Together, these thus provide a rotation that is indistinguishable from the rotation obtained from the Ulukışla basin and the southern Kırşehir Block.

The ~10° rotation difference between the Cretaceous sites close to the EFZ (Fig. 1 arrow with reference 3; (Çinku et al., 2016)) and the Eocene sites towards the east may be explained in two ways. Either, they may represent a small rotation associated with drag folding along the left-lateral EFZ. Alternatively, they may indicate a ~10° rotation occurring between

the Late Cretaceous and the Eocene, when the Tauride rocks were still connected to the downgoing African plate. Between ~80 and ~50 Ma, Africa did experience a 10° ccw rotation. In absence of a detailed structural model for the eastern Taurides showing from which nappes the paleomagnetic data were obtained, and when those nappes were incorporated in the Tauride fold-and-thrust belt, we cannot determine which of these two solutions is the more likely. We interpret that the Taurides have





undergone a coherent 30°ccw rotation since Oligocene time, and that the shearing along the EFZ did not lead to strong regional extra rotations.

Oligocene sediments overlying the Eastern Tauride fold-and-thrust belt in our Aladağ locality reveal a post-folding magnetization revealing a 17±7°ccw rotation. Since the timing of its remagnetization cannot be constrained and may well

have occurred during the rotation phase, this sheds no light on the timing of the eastern Tauride rotation. Our Miocene site Gürün shows a counter-clockwise rotation of 16±11°, and the Kepezdag and Yamadag localities of Gürsoy et al., (2011) obtained from middle Miocene (13-15 Ma) lavas show 26±20° and 28±6° ccw, respectively (**Fig. 1**, arrows with reference 6). Taken together, those data may suggest that the eastern Tauride rotations occurred in Middle Miocene or younger time, but we note that these Miocene lavas were sampled in the close vicinity of Middle Miocene and younger left-lateral strike-

slip faults (Westaway & Arger, 2001; Kaymakci et al., 2010), and may thus not be representative for the timing of regional rotation. Three sites (**Fig. 1**; arrows with references 8, 11) from middle and upper Miocene sediments in the Adana basin (Çinku et al., 2016; Lucifora et al., 2012) yield a declination of 354.3±3.8° (n=134, K=14.4), suggesting a minor rotation of 5.7±3.8°, suggesting a pre-middle Miocene rotation. We tentatively suggest that the regional rotation of the eastern Taurides occurred sometime between the Eocene and middle Miocene, and was further locally modified by Miocene left-lateral strike-

slip faults.

Overall, the sense and magnitude of the Central and Eastern Tauride rotations, those obtained from the stratigraphy of the Ulukışla basin and those previously reported from the southern part of the Kırşehir Block (Lefebvre et al., 2013), suggest that the region rotated as a more or less coherent block, which has undergone a counter-clockwise rotation of ~30°, (largely) sometime in Oligocene or Early Miocene time.

### 5.3 Rotations in the Sivas basin, and the boundary of the south-east Anatolian counter-clockwise rotating domain

Our results from the Sivas basin show that extensive sampling was required to obtain rotation patterns, as only ~30% of samples and sites yielded meaningful paleomagnetic results. This is largely caused by (true or suspected) post-folding remagnetization, particularly to the north of the Deliler-Tecer Fault Zone (DTFZ). Nevertheless, our results lead us to the

following first-order interpretation of rotations in the Sivas basin.

We subdivide the Sivas region into three domains (**Fig. 8**). These are 1) the area exposing Eocene and Oligocene sediments to the south of, and in the footwall of the top-to-the-south DTFZ, 2) the area exposing Paleogene sediments immediately north, and in the hangingwall of this thrust zone, and 3) the Miocene and younger sediments occupying the northernmost part of the basin. The rotations in the Oligocene corridor of the southern domain (**Fig. 3**) are constrained by – from E to W -

the Sincan, Inkonak and Yeniköy localities, which reveal a combined counter-clockwise rotation of 48.1±2.9° (n=191, K=15.8). Also the Middle Miocene Gemerek locality (Krijgsman et al., 1996) yielded such high rotations (49.9±3.8°, Table 1). Interestingly, these rotations are ~15-20° more ccw than those recorded in the Eastern Tauride fold-thrust belt. This difference may be interpreted in different ways. On the one hand, it may suggest that the Taurides underwent a clockwise rotation between the Eocene and Oligocene of 15-20°, followed by a 50° rotation after deposition of the middle Miocene





Gemerek section. This, however, is inconsistent with the directions obtained from Oligocene sediments in the Ulukışla basin or Miocene sediments overlying the Taurides (**Fig. 2**). We thus assume that the area to the south of the DTFZ was part of the Eastern Tauride rotating domain and tentatively ascribing the excess ~15° rotation to the local deformation in the footwall of the DTFZ, e.g. introduced by a sinistral component of the fault zone as postulated by Yilmaz and Yilmaz, (2006).

A distinct break in rotation sense occurs across the DTFZ (**Fig. 3, Fig. 8**), north of which rotations are variable but generally clockwise, rotating as much as ~30° in the central and northern part of the basin (Gürlevik, Ulaş, Sivas, Kemah localities). Explaining the variable clockwise rotations is not straightforward, but well-documented intense deformation, in part associated with intense local salt tectonics (Kergaravat et al., 2016; Pichat et al., 2016; Poisson et al., 2016; Ribes et al., 2015) may provide an explanation. We note, however, that one Oligocene locality (Güllük) to the north of the DTFZ reveals

the counter-clockwise rotations characteristic for the area to its south. Farther to the north, in the eastern Kırşehir block, or the eastern Pontides north of the North Anatolian Fault Zone, no major counter-clockwise rotations were reported. The northeastern Kırşehir block (the AYB block of Lefebvre et al. (2013) experienced 18.4±6.0° clockwise rotation measured in Upper Cretaceous granitoids. The eastern Pontides experienced only minor rotations since Late Cretaceous-Eocene time, and a compilation of existing data (Baydemir, 1990; Channell et al., 1996; Kissel et al., 2003; Meijers et al., 2010; Orbay and

Bayburdi, 1979; van der Voo, 1968).

We therefore suggest that the boundary of the coherently counter-clockwise rotating domain of the Eastern Taurides most likely coincides with the Deliler-Tecer Fault Zone (DTFZ) (**Fig. 8**). Previously, Lefebvre et al. (2013) suggested that the counter-clockwise rotation of the southern Kırşehir block was bounded in the north along the top-to-the-south Savcılı Thrust Zone (STZ), below which deformed and folded sediments are found similar in age and facies as those below the DTFZ.

Given this similarity, and the coherence in rotation direction, amount, and timing, we infer that the counter-clockwise rotating domain of central and southern Anatolia, comprising the southern Kırşehir Block, the Ulukışla basin, the Bolkar mountains, the eastern Taurides, and the southern part of the Sivas basin, was bounded to the north by a thrust fault zone comprising the STZ in the west, and the DTFZ in the east.

Finally, the eastern Taurides appear to have rotated coherently with the southern Kırşehir block and the Ulukışla basin, but

we note that they were displaced relative to the latter along the EFZ (**Fig. 8**). This fault is controlled by the eastern margin of the Kırşehir Block, and its left-lateral displacement requires that the Sivas basin underwent 60-70 km more N-S Eocene-Early Miocene convergence than Central Anatolia. This excess convergence is likely responsible for the much stronger deformation, and probably the more disperse rotation patterns, associated with the structural growth of the Sivas fold-thrust belt.

**6 Conclusion**

In this paper, we provide a large set of new paleomagnetic data from central southern and central eastern Anatolia to aid kinematic restoration of the Anatolian Orogen. We aimed to identify the timing, amount, and regional coherence of rotating



blocks in central and southern Anatolia. Our main findings can be summarized as follows:

1. The Ulukışla basin underwent a regional counter-clockwise rotation of ~32.5±2.2° in Oligocene and Miocene times, comparable with the amount and sense of rotation (28-35°) previously reported for the southern part of the Kırşehir block. This rotation phase is contemporaneous with the activity of the STZ as a contractional structure and onset of the major left-lateral EFZ, and possibly with similar structures within the eastern Taurides (e.g. Malatya Fault). Deviations from this regionally consistent pattern are found in the southern and south-eastern part of the basin, where local rotations strongly deviating from this average, owing to vicinity to major tectonic structures, such as the EFZ.

2. We find ~17 counter-clockwise rotation in Eocene, but remagnetized, sedimentary rocks overlying the Central Tauride fold-thrust belt in the Bolkar mountains. This suggests that the Bolkar mountains were part of the counter-clockwise rotating domain as opposed to clockwise rotations found in the western Central Taurides. The total amount of counter-clockwise rotation since the Eocene cannot be determined due to remagnetization, but absence of a major Oligocene or younger fault between the Ulukışla basin and the Bolkar mountains lead us to include these in the counter-clockwise rotating south Anatolian domain.

3. Counter-clockwise rotations were previously reported from the Eastern Taurides. These show a comparable ~30° ccw rotation since the Eocene. Larger ccw rotations are reported close to the EFZ, which dissects the Taurides in its central and eastern parts, and plays a major role in the growth of the Sivas fold-and-thrust belt. Our new paleomagnetic data from the Sivas fold-and-thrust belt, reveals that on average ~48° counter-clockwise rotations can be traced into the footwall of the DTFZ. In the hanging wall of this thrust, paleomagnetic data quality is generally low, but successful sites show consistently minor (7-20°) clockwise rotations.

4. We conclude that the southern Kırşehir Block, Ulukışla basin, Bolkar mountains, eastern Taurides, and the southern part of the Sivas basin were part of one coherently counter-clockwise rotating domain that experienced ~30° rotation in Oligocene or earliest Miocene time. Structural constraints suggest that this domain was bounded in the north by the STZ and DTFZ and in the south by the African-Arabian trench. To the west, the boundary is diffuse and requires future study.

**Author contribution**

DG and DJJvH designed the study. DG, MÖ carried out the bulk of the sampling, with support from DJJvH, IC, MK, CL. IC and MK helped with acquiring paleomagnetic data. AC provided biostratigraphic constraints. DG, DJJvH, MÖ, CL, interpreted the data.





**Acknowledgments**

DG, DJJvH were supported by ERC-Starting Grant 306810. DJJvH acknowledges NWO Vidi grant 864.11.004. Pinar
Ertepinar and Nur Güneli are thanked for field assistance and logistical support. W. Krijgsman is thanked for providing the
original data files of a previously published dataset. DG and DJJvH prepared the manuscript. The authors declare that they
have no conflict of interest.

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





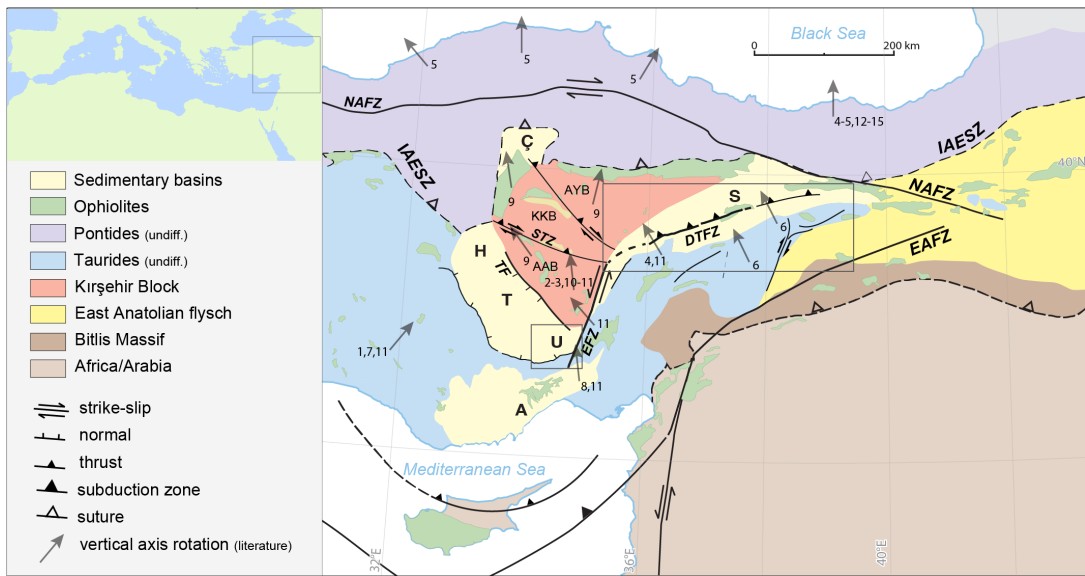

**Figure 1:** Tectonic units, associated suture zones, main fault zones within the Anatolian microplate. Available vertical axis rotations are shown as arrows, corresponding references are:(**1:** Kissel et al., 1993; **2:** Platzman et al., 1998; **3:** Piper et al., 2002; **4:** Kissel et al., 2003; **5:**Meijers et al., 2010a; **6:** Gürsoy et al., 2011; **7:** Meijers et al., 2011; **8:** Lucifora et al., 2012; **9:** Lefebvre et al., 2013; **10:** Piper et al., 2013; **11:** Çinku et al., 2016 **12:** Baydemir, 1990; **13:** Channell et al., 1996; **14:** Orbay and Bayburdi, 1979; **15**: van der Voo, 1968). Location of the sedimentary basins mentioned: Ulukışla (U), Sivas (S), Haymana (U), Tuzgölü (T), Çankırı (Ç), Adana (A). Major faults: Ecemiş Fault zone (EFZ), Deliler-Tecer Fault Zone (DTFZ), Savcılı Thrust Zone (STZ), Tuzgölü Fault (TF), North Anatolian Fault (NAFZ), East Anatolian Fault Zone (EAFZ). The Kırşehir Block consists of the Ağacören-Yozgat block (AYB), the Kırşehir–Kırıkkale block (KKB), and the Ağacören–Avanos block (AAB). Note that the U and S basins are offset along the EFZ. Map insets of the study regions between the Kırşehir Block and the Taurides, and the Kırşehir Block, the Eastern Pontides and Taurides are shown in **Figs. 2 and 3.**




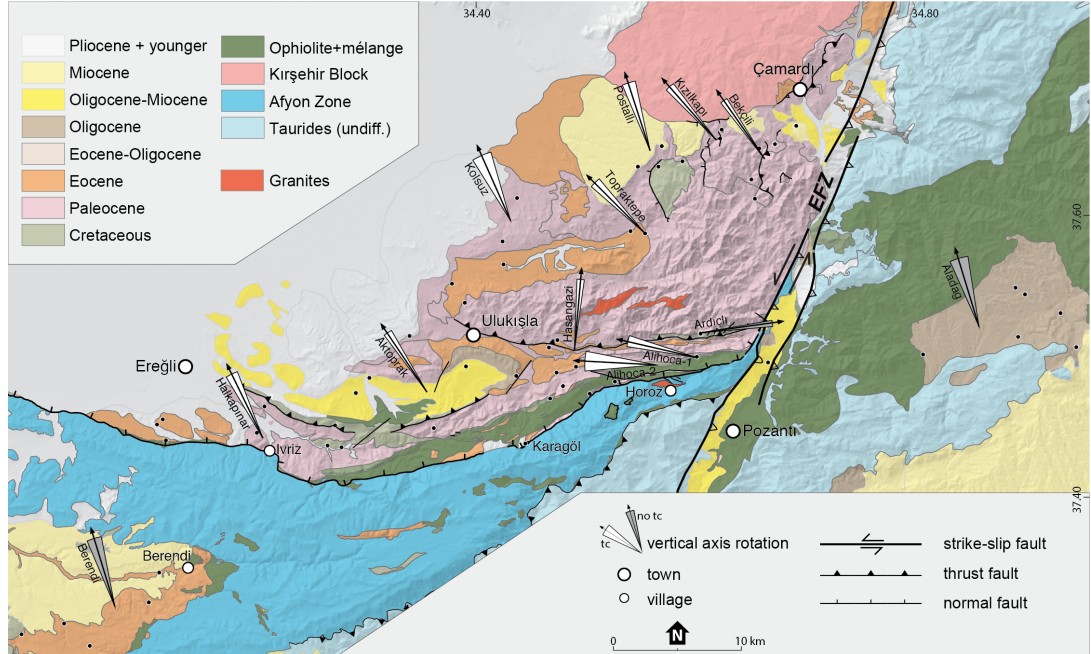

**Figure 2:** Geological map of the Ulukışla basin (modified after Gürer et al., 2016), sampling sites (black dots), localities, vertical axis rotations are denoted as arrows with their 95% error envelope ($\Delta D_x$).





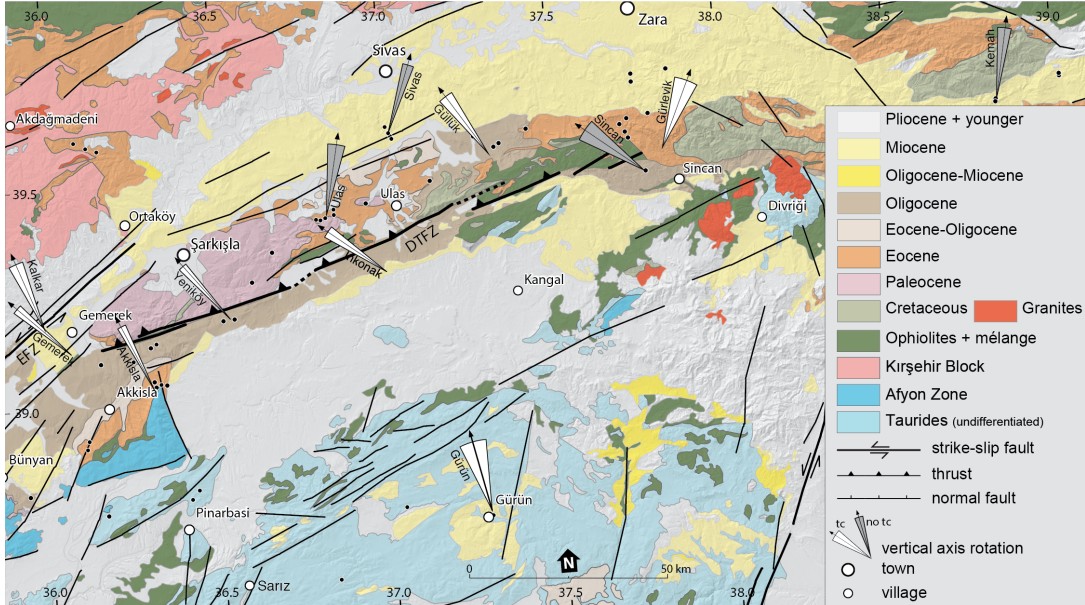

**Figure 3:** Geological map of the Sivas Basin (modified from MTA, 2002), symbols as in caption to figure 2.



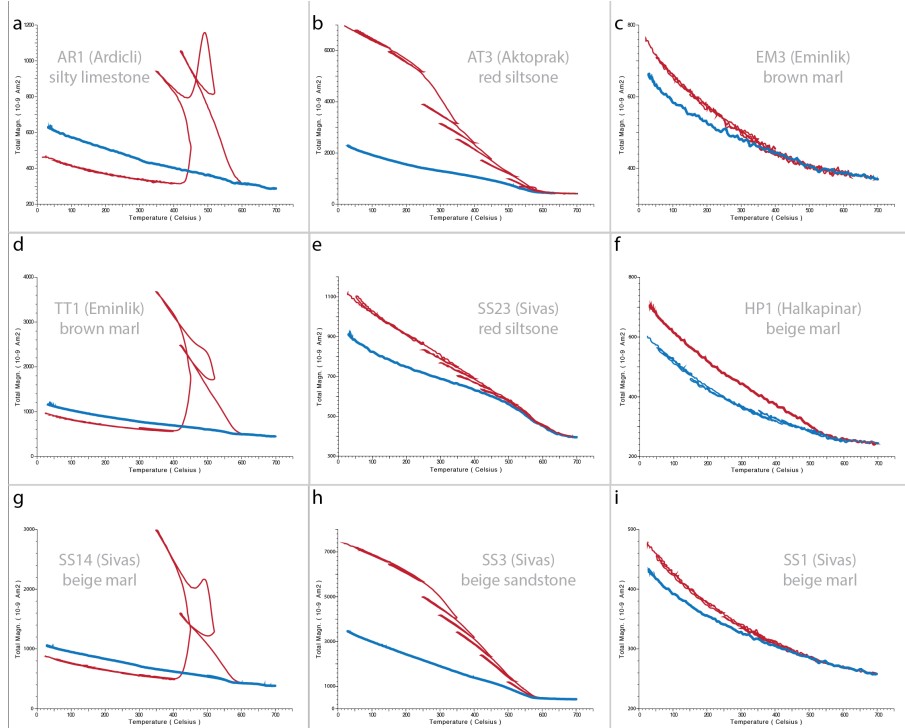

**Figure 4:** Magnetic carriers identified by their characteristic thermomagnetic curves generated with the stepwise heating protocol (Mullender et al., 1993) for representative samples. Heating is represented by red line. The final cooling segment is indicated with thicker blue line. A noisy appearance is indicative of a weak magnetic signal. See for text for explanation of the thermomagnetic behaviour.





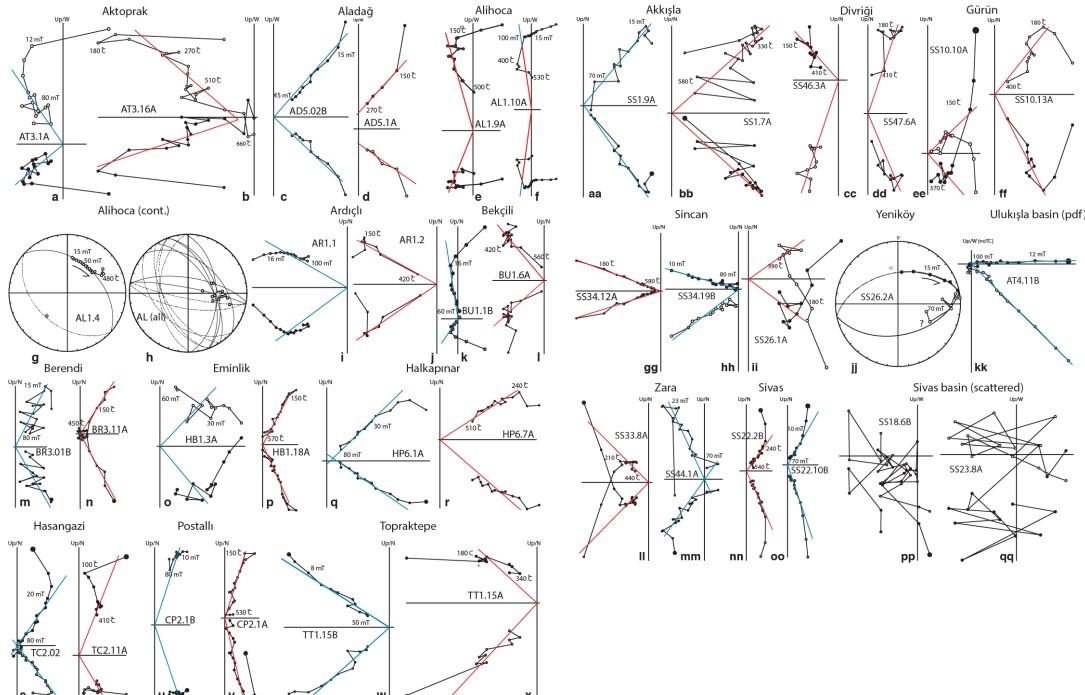

**Figure 5:** Zijderveld diagrams (Zijderveld, 1967) of representative samples demagnetized using thermal (red lines, TH) and alternating field (blue lines, AF) demagnetization shown in in situ (noTC) or tectonic (all other diagrams) coordinates. The solid and open dots represent projections on the horizontal and vertical planes, respectively. Great circle plots of g, h, and jj, use the technique of McFadden and McElhinny (1988). Demagnetization step values are in °C or in mT. See text for further explanation.



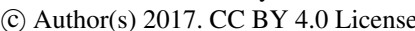

**Figure 6:** Locality results from sedimentary basins (Ulukışla and Sivas basins, and basins overlying the Tauride fold-and-thrust belt). Equal area projections of ChRM directions and their means with associated error ellipses ($\Delta D_x$, $\Delta I_x$) according to (Deenen et al., 2011), either before (orange; NoTC) or after tectonic correction (blue; TC). Rejected directions (after 45° cut-off) are displayed in grey, positive (negative) inclinations are shown as solid (open) circles.




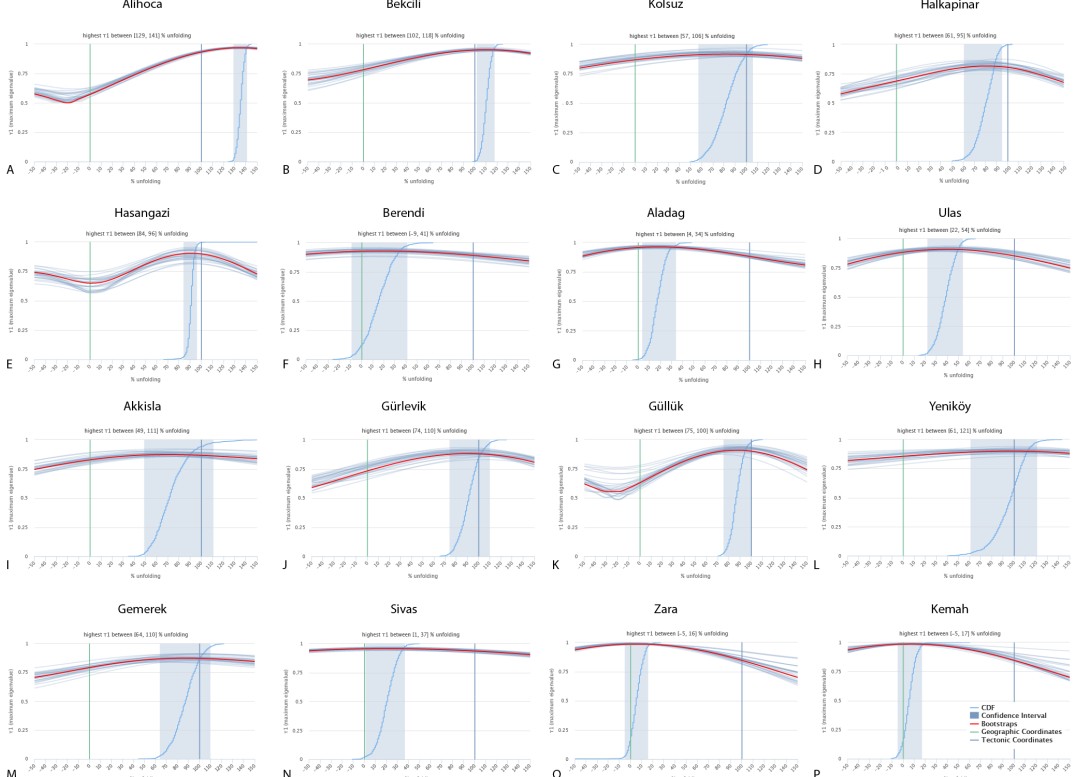

**Figure 7:** Representative fold tests after Tauxe et al. (2010) for localities from the Ulukışla and Sivas basins.





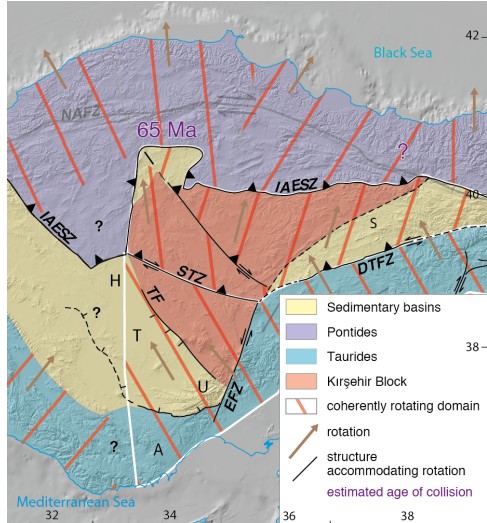

**Figure 8:** Coherently rotating domains within the Anatolian orogen and tectonic structures accommodating differential rotations. NAFZ - North Anatolian Fault Zone, IAESZ – Izmir-Ankara-Erzincan Suture Zone, STZ – Savcılı Thrust Zone, DTFZ – Deliler-Tecer Fault Zone, TF – Tuzgölü Fault, EFZ – Ecemiş Fault zone, basins, U – Ulukışla , T- Tuzgölü, Haymana, Adana. Orange hatching indicates the amount of vertical axis rotation of each domain, whereby the hatching is rotated from N-S according to the paleomagnetic results summarized in **Table 1**.



| Site | Age | lat (°) | long (°) | Na | Ni | N45 | in situ D | DDx | I | DIx | tilt corrected D | DDx | I | DIx | K | A95 | A95min | A95max | Rot | DRot | sense |
|---|---|---|---|---|---|---|---|---|---|---|---|---|---|---|---|---|---|---|---|---|---|
| **ULUKISLA BASIN** | | | | | | | | | | | | | | | | | | | | | |
| **Alihoca-1** | **Coniacian-Maastrichtian** | **37.5** | **34.7** | **111** | **47** | **47** | | | | | **283.1** | **4.4** | **40.3** | **5.6** | **27.4** | **4.0** | **2.6** | **7.3** | negative reversal/fold test | | |
| AL1 | | 37.505 | 34.733 | 24 | 24 | 24 | 19.4 | 3.7 | -23.8 | 6.3 | 91.5 | 4.8 | -47.4 | 5.0 | 50.5 | 4.2 | 3.4 | 11.1 | | | |
| *AL3* | | *37.485* | *34.697* | *46* | | | *no result* | | | | | | | | | | | | | | |
| *AL4 (pdf)* | | *37.471* | *34.659* | *16* | | | *355.6* | *9.0* | *60.9* | *5.7* | | | | | | | | | | | |
| AR2 | | 37.513 | 34.822 | 25 | 23 | 23 | 217.7 | 4.5 | 42.6 | 5.4 | 292.8 | 3.7 | 31.9 | 5.5 | 75.5 | 3.5 | 3.4 | 11.4 | | | |
| **Alihoca-2** | **Selandian (L. Paleocene)** | **37.5** | **34.7** | **19** | **15** | **15** | | | | | **272.9** | **8.2** | **44.0** | **9.4** | **27.9** | **7.4** | **4.1** | **14.9** | **87.1** | **8.2** | **ccw** |
| AL2 | | 37.506 | 34.734 | 19 | 15 | 15 | 22.8 | 5.7 | -25.5 | 9.5 | 92.9 | 8.2 | -44.0 | 9.4 | 27.9 | 7.4 | 4.1 | 14.9 | | | |
| **Ardicli** | **Campanian-Maastrichtian** | **37.6** | **34.8** | **34** | **23** | **23** | **82.4** | **2.4** | **47.1** | **2.5** | **45.9** | **2.6** | **29.7** | **4.1** | **146.7** | **2.5** | **3.4** | **11.4** | **82.4** | **2.4** | **cw** |
| AR1 | | 37.541 | 34.828 | 19 | 15 | 15 | 261.5 | 2.8 | -47.0 | 3.0 | 223.2 | 2.6 | -30.4 | 4.1 | 241.3 | 2.5 | 4.1 | 14.9 | | | |
| AR3 | | 37.540 | 34.828 | 15 | 8 | 8 | 264.1 | 5.2 | -47.3 | 5.4 | 230.7 | 4.7 | -28.2 | 7.6 | 149.6 | 4.5 | 5.2 | 22.1 | | | |
| **Bekcili** | **Thanetian (L. Paleocene)** | **37.8** | **34.9** | **139** | **70** | **70** | | | | | **324.6** | **2.9** | **37.7** | **3.9** | **39.9** | **2.7** | **2.2** | **5.6** | **35.4** | **2.9** | **ccw** |
| BU1 | | 37.767 | 34.945 | 23 | 5 | 5 | 297.8 | 13.3 | 36.3 | 18.4 | 329.8 | 19.3 | 41.5 | 23.6 | 19.9 | 17.6 | 6.3 | 29.7 | | | |
| BU2 | | 37.766 | 34.946 | 33 | 6 | 6 | 147.4 | 13.7 | 0.1 | 27.4 | 154.1 | 14.6 | -25.8 | 24.4 | 23.2 | 14.2 | 5.9 | 26.5 | | | |
| YK1 | | 37.701 | 34.914 | 14 | 13 | 13 | 162.6 | 4.1 | -16.3 | 7.6 | 147.8 | 5.1 | -36.5 | 7.1 | 75.4 | 4.8 | 4.3 | 16.3 | | | |
| YK2 | | 37.724 | 34.910 | 13 | 13 | 13 | 164.6 | 5.1 | -19.5 | 9.2 | 147.6 | 6.6 | -40.0 | 8.4 | 47.0 | 6.1 | 4.3 | 16.3 | | | |
| YK3 | | 37.761 | 34.922 | 29 | 20 | 20 | 13.9 | 5.2 | 57.6 | 3.8 | 315.8 | 3.6 | 44.7 | 4.0 | 105.3 | 3.2 | 3.6 | 12.4 | | | |
| YK4 | | 37.766 | 34.146 | 27 | 13 | 13 | 309.4 | 5.3 | 14.0 | 10.1 | 324.3 | 5.5 | 28.7 | 8.7 | 62.7 | 5.3 | 4.3 | 16.3 | | | |
| **Kizilkapi** | **Thanetian (L. Paleocene)** | **37.8** | **34.8** | **115** | **55** | **54** | **134.5** | **5.4** | **-46.5** | **5.7** | **193.2** | **7.2** | **-64.2** | **3.9** | **16.3** | **5.0** | **2.5** | **6.7** | **13.2** | **7.2** | **cw** |
| CP1 | | 37.785 | 34.866 | 16 | 12 | 12 | 130.2 | 10.0 | -36.8 | 13.6 | 172.7 | 19.4 | -61.2 | 12.0 | 10.3 | 14.2 | 4.4 | 17.1 | | | |
| *PC1* | | *37.742* | *34.769* | *31* | *26* | *23* | *52.5* | *9.5* | *-46.7* | *10.1* | *357.1* | *15.3* | *-64.8* | *8.0* | *9.5* | *10.4* | *3.4* | *11.4* | | | |
| PC2 | | 37.774 | 34.809 | 22 | 19 | 19 | 114.9 | 5.6 | -49.6 | 5.4 | 172.8 | 11.8 | -68.8 | 5.0 | 22.8 | 7.2 | 3.7 | 12.8 | | | |
| *PC3* | | *37.774* | *34.864* | *20* | | | *no result* | | | | | | | | | | | | | | |
| PC4 | | 37.783 | 34.867 | 26 | 24 | 24 | 154.1 | 5.2 | -46.0 | 5.7 | 209.5 | 7.4 | -60.7 | 4.7 | 29.9 | 5.5 | 3.4 | 11.1 | | | |
| **Kolsuz** | **Paleocene** | **37.7** | **34.5** | **48** | **18** | **18** | | | | | **335.8** | **7.8** | **32.5** | **11.7** | **22.4** | **7.5** | **3.8** | **13.3** | **24.2** | **7.8** | **ccw** |
| KL1 | | 37.655 | 34.531 | 28 | 8 | 8 | 157.6 | 6.6 | -23.8 | 11.2 | 163.6 | 7.1 | -36.0 | 9.9 | 70.1 | 6.7 | 5.2 | 22.1 | | | |
| EM3 | | 37.707 | 34.554 | 20 | 10 | 10 | 132.3 | 15.1 | -45.7 | 16.5 | 149.7 | 1.5 | -29.2 | 20.2 | 16.4 | 12.3 | 4.8 | 19.2 | | | |
| **Halkapinar** | **Eocene** | **37.4** | **34.3** | **196** | **69** | **68** | | | | | **334.8** | **4.9** | **33.4** | **7.3** | **14.4** | **4.7** | **2.2** | **5.7** | **25.2** | **4.9** | **ccw** |
| *YL1 (remagnetized)* | | *37.422* | *34.019* | *11* | *5* | *5* | *175.5* | *7.2* | *-53.1* | *6.2* | *176.7* | *4.5* | *-25.1* | *7.5* | *312.6* | *4.3* | *6.3* | *29.7* | | | |
| HP1 | | 37.437 | 34.274 | 31 | 24 | 24 | 158.0 | 8.9 | 44.6 | 10.1 | 160.9 | 6.9 | -36.6 | 9.4 | 22.1 | 6.4 | 3.4 | 11.1 | | | |
| HP2 | | 37.411 | 34.283 | 21 | 6 | 6 | 305.5 | 49.5 | 73.2 | 14.5 | 330.4 | 15.6 | 31.5 | 23.7 | 21.1 | 14.9 | 5.9 | 26.5 | | | |
| HP3 | | 37.461 | 34.168 | 26 | 16 | 15 | 168.8 | 8.6 | -3.8 | 17.0 | 142.0 | 12.0 | -50.5 | 11.4 | 14.8 | 10.3 | 4.1 | 14.9 | | | |
| *HP4 (remagnetized)* | | *37.430* | *34.156* | *14* | *14* | *7* | *156.1* | *2.8* | *-65.4* | *1.4* | *182.3* | *1.3* | *9.2* | *2.5* | *2180.3* | *1.3* | *5.5* | *24.1* | | | |
| *HP5* | | *37.413* | *34.262* | *13* | *13* | | *no result* | | | | | | | | | | | | | | |
| HP6 | | 37.432 | 34.423 | 20 | 11 | 11 | 147.7 | 8.5 | -0.8 | 16.9 | 153.2 | 10.5 | -38.8 | 13.7 | 23.1 | 9.7 | 4.6 | 18.1 | | | |
| *KG1 (pdf)* | | *37.413* | *34.564* | *30* | *20* | *20* | *0.3* | *4.9* | *56.6* | *3.7* | *341.2* | *2.2* | *16.4* | *4.1* | *71.9* | *3.9* | *3.6* | *12.4* | | | |
| KG2 | | 37.412 | 34.561 | 30 | 30 | 12 | 182.6 | 15.5 | -70.1 | 6.1 | 168.9 | 6.4 | -35.4 | 9.0 | 53.4 | 6.0 | 4.4 | 17.1 | | | |
| **Eminlik** | **Eocene** | **37.6** | **34.6** | **70** | | | | | | | | | | | | | | | | | |
| *EM1 (pdf)* | | *37.665* | *34.725* | *17* | | | *355.2* | *5.1* | *52.6* | *4.5* | | | | | | | | | | | |
| *EM2 (remagnetized)* | | *37.628* | *34.532* | *16* | *8* | *8* | *179.4* | *12.5* | *-56.1* | *9.6* | *162.5* | *8.2* | *-7.5* | *16.2* | *-72.3* | *8.2* | *5.2* | *22.1* | | | |
| *EM4 (remagnetized)* | | *37.606* | *34.454* | *23* | *23* | *23* | *191.7* | *11.7* | *-67.4* | *5.4* | *179.2* | *3.7* | *-14.7* | *7.0* | *68.0* | *3.7* | *3.4* | *11.4* | | | |
| *EM5 (pdf)* | | *37.582* | *34.473* | *14* | | | *352.3* | *9.1* | *60.4* | *5.9* | | | | | | | | | | | |
| **Hasangazi** | **Eocene** | **37.5** | **34.6** | **110** | **41** | **38** | | | | | **4.9** | **2.8** | **31.4** | **4.2** | **77.8** | **2.6** | **2.8** | **8.3** | **4.9** | **2.8** | **cw** |
| *HG1 (pdf)* | | *37.536* | *34.612* | *10* | *9* | *9* | *356.3* | *4.7* | *60.6* | *3.0* | *90.3* | *45.8* | *85.8* | *3.0* | *75.9* | *5.9* | *5.0* | *20.5* | | | |
| HG2 | | 37.527 | 34.599 | 16 | 16 | 13 | 183.2 | 4.4 | 21.4 | 7.8 | 185.1 | 5.1 | -37.7 | 6.9 | 75.9 | 4.8 | 4.3 | 16.3 | | | |
| *TC1* | | *37.497* | *34.610* | *22* | *22* | | *no result* | | | | | | | | | | | | | | |
| TC2 | | 37.503 | 34.639 | 27 | 25 | 25 | 240.6 | 4.9 | -58.6 | 3.4 | 184.8 | 3.2 | -28.2 | 5.2 | 87.9 | 3.1 | 3.3 | 10.8 | | | |
| *GM1* | | *37.466* | *34.599* | *16* | | | *no result* | | | | | | | | | | | | | | |
| *GM2 (pdf)* | | *37.480* | *34.619* | *19* | *7* | *7* | *2.7* | *11.9* | *57* | *89* | *352.2* | *4.1* | *2* | *8.1* | *222.8* | *4.1* | *5.5* | *24.1* | | | |
| **Topraktepe** | **Middle Eocene** | **37.7** | **34.8** | **30** | **29** | **28** | | | | | **315.5** | **4.1** | **42.8** | **4.9** | **53.4** | **3.8** | **3.2** | **10.0** | **44.5** | **4.1** | **ccw** |
| TT1 | | 37.660 | 34.750 | 30 | 29 | 28 | 4.6 | 3.3 | 26.7 | 5.4 | 315.5 | 4.1 | 42.8 | 4.9 | 53.4 | 3.8 | 3.2 | 10.0 | | | |
| **Aktoprak** | **Chattian (L. Oligocene)** | **37.5** | **34.4** | **398** | **148** | **144** | | | | | **326.3** | **3.8** | **46.6** | **4.0** | **13.5** | **3.3** | **1.6** | **3.6** | **33.7** | **3.8** | **ccw** |
| *AT1* | | *37.493* | *34.549* | *15* | | | *no result* | | | | | | | | | | | | | | |
| *AT2* | | *37.506* | *34.473* | *22* | | | *no result* | | | | | | | | | | | | | | |
| AT3 | | 37.494 | 34.327 | 27 | 27 | 26 | 48.1 | 12.1 | -60.1 | 7.9 | 150.3 | 8.3 | -34.6 | 11.9 | 14.0 | 7.9 | 3.3 | 10.5 | | | |
| *AT4 (pdf)* | | *37.476* | *34.306* | *21* | *20* | *20* | *4.9* | *3.0* | *57.0* | *3.1* | *10.0* | *1.5* | *2.4* | *3.1* | *454.9* | *1.5* | *3.6* | *12.4* | | | |
| Kurtulmus Tepe (Meijers et al., 2016) | | 37.522 | 34.475 | 313 | 129 | 121 | 338 | 2.7 | 18.5 | 5.0 | 326.4 | 4.3 | 49.6 | 4.2 | 12.8 | 3.7 | 1.8 | 4.0 | | | |
| **Postalli** | **Miocene** | **37.8** | **34.8** | **40** | **27** | **27** | | | | | **342.3** | **5.8** | **36** | **8.1** | **27.2** | **5.4** | **3.2** | **10.3** | **17.7** | **5.4** | **ccw** |
| CP2 | | 37.757 | 34.772 | 16 | 12 | 12 | 144.3 | 21.1 | -68.6 | 9.0 | 163.6 | 10.2 | -43.3 | 12.0 | 23.0 | 9.2 | 4.4 | 17.1 | | | |
| CP3 | | 37.742 | 34.738 | 24 | 15 | 15 | 359.5 | 9.6 | 51.8 | 8.8 | 341.5 | 6.7 | 30.2 | 10.5 | 35.8 | 6.5 | 4.1 | 14.9 | | | |
| **Hacibekirli** | **Miocene** | **37.5** | **34.4** | **24** | | | | | | | | | | | | | | | | | remagnetized? |
| HB1 | | 37.544 | 34.388 | 24 | 18 | 18 | 182.8 | 4.1 | -52.6 | 3.6 | 164.8 | 3.1 | -34.1 | 4.5 | 137.2 | 3.0 | 3.8 | 13.3 | | | |
| **Burc** | **Miocene** | | | | | | | | | | | | | | | | | | | | |
| *BU3* | | *37.788* | *34.979* | *17* | | | *no result* | | | | | | | | | | | | | | |
| | | | | | | | | | | | | | | | | | | | | | |
| **Ulukisla_all_directions** | **Paleocene-Oligocene** | **37.6** | **34.7** | **334** | **326** | | | | | | **327.7** | **2.2** | **40.5** | **2.7** | **16.4** | **2.0** | **1.2** | **2.1** | **32.3** | **2.2** | **ccw** |
| **Ulukisla_all_localities** | **Paleocene-Oligocene** | **37.6** | **34.7** | **5** | **5** | | | | | | **327.8** | **9.0** | **38.8** | **11.8** | **84.5** | **8.4** | | | **32.2** | **9.0** | **ccw** |





| Site | Age | lat (°) | long (°) | Na | Ni | N45 | in situ | | | | tilt corrected | | | | K | A95 | A95min | A95max | Rot | DRot | sense |
|---|---|---|---|---|---|---|---|---|---|---|---|---|---|---|---|---|---|---|---|---|---|
| | | | | | | | D | DD_x | I | DI_x | D | DD_x | I | DI_x | | | | | | | |
| **TAURIDE FTB and OVERLYING BASINS** | | | | | | | | | | | | | | | | | | | | | |
| **Berendi** | Thanetian (L. Paleocene)-Ypresian (E.Eocene) | 37.2 | 33.9 | 58 | 40 | 39 | 343.1 | 6.2 | 53.7 | 5.3 | | | | | 21.0 | 5.1 | 2.8 | 8.2 | 16.9 | 6.2 | ccw |
| BR1 | | | | 12 | 6 | 6 | 161.0 | 16.1 | -43.0 | 19.0 | 136.6 | 11.6 | -34.0 | 16.8 | 38.4 | 10.9 | 5.9 | 26.5 | | | |
| BR2 | | | | 10 | 6 | 6 | 176.6 | 8.8 | -65.5 | 4.5 | 165.7 | 6.0 | -51.9 | 5.4 | 174.6 | 5.1 | 5.9 | 26.5 | | | |
| BR3 | | | | 14 | 10 | 10 | 157.6 | 6.3 | -57.6 | 4.6 | 160.0 | 4.2 | -41.7 | 5.1 | 159.9 | 3.8 | 4.8 | 19.2 | | | |
| BR4 | | | | 13 | 12 | 12 | 163.0 | 6.4 | -52.6 | 5.6 | 184.0 | 5.8 | -43.3 | 6.8 | 68.9 | 5.3 | 4.4 | 17.1 | | | |
| *BR5 (pdf)* | | | | *9* | *9* | *9* | *0.0* | *4.3* | *52.9* | *3.8* | *354.8* | *3.6* | *45.2* | *4.0* | *256.2* | *3.2* | *5.0* | *20.5* | | | |
| *BR6* | | | | | | | *no result* | | | | | | | | | | | | | | |
| **Aladag** | Oligocene | 37.5 | 35.3 | 56 | 20 | 20 | 343.4 | 7.3 | 52.9 | 6.3 | | | | | 29.9 | 6.1 | 3.6 | 12.4 | 16.6 | 7.3 | ccw |
| *AD1* | | | | *16* | | | *no result* | | | | | | | | | | | | | | |
| *AD3 (norm)* | | | | *18* | | | *no result* | | | | | | | | | | | | | | |
| AD4 | | | | 13 | 13 | 13 | 341.6 | 10.5 | 57.0 | 7.8 | 351.6 | 5.3 | 28.7 | 8.4 | 67.9 | 5.1 | 4.3 | 16.3 | | | |
| AD5 | | | | 9 | 7 | 7 | 345.9 | 7.4 | 45.5 | 8.1 | 309.2 | 5.8 | 34.7 | 8.3 | 121.7 | 5.5 | 5.5 | 24.1 | | | |
| **Sariz** | L.Cretaceous-Eocene | 38.7 | 36.7 | 63 | | | | | | | | | | | | | | | | | | |
| *SS06* | | *38.592* | *36.834* | *21* | | | | | | | | | | | | | | | | | |
| *SS07* | | *38.809* | *36.430* | *17* | | | | | | | | | | | | | | | | | |
| *SS08* | | *38.788* | *36.409* | *16* | | | | | | | | | | | | | | | | | |
| *SS09* | | *38.752* | *37.035* | *9* | | | | | | | | | | | | | | | | | |
| **Gürün** | Miocene | 38.7 | 37.3 | 11 | 9 | 9 | 213.1 | 12.0 | 53.1 | 10.3 | 343.9 | 10.9 | 42.9 | 12.9 | 32.2 | 9.9 | 5.2 | 22.1 | 16.1 | 10.9 | ccw |
| SS10 | | 38.724 | 37.286 | 11 | 8 | 8 | 146.9 | 12.0 | -53.1 | 10.3 | 163.9 | 10.9 | -42.9 | 12.9 | 32.2 | 9.9 | 5.2 | 22.1 | | | |
| **SIVAS BASIN** | | | | | | | | | | | | | | | | | | | | | |
| **Ulas** | Paleocene-Eocene | 39.4 | 36.8 | 207 | 32 | 32 | 7.9 | 7.2 | 43.0 | 8.5 | | | | | 12.4 | 7.7 | 3.0 | 9.4 | 7.9 | 7.2 | cw |
| *SS11* | | *39.466* | *37.006* | *20* | | | | | | | | | | | | | | | | | |
| SS17 | | 39.440 | 36.856 | 32 | 12 | 12 | 187.9 | 14.5 | -62.3 | 8.6 | 222.2 | 6.5 | -28 | 10.5 | 48.3 | 6.3 | 4.4 | 17.1 | | | |
| *SS18* | | *39.428* | *36.856* | *31* | | | | | | | | | | | | | | | | | |
| SS19 | | 39.421 | 36.830 | 21 | 15 | 15 | 190.9 | 6.7 | -33.2 | 9.9 | 174.5 | 6.2 | -32 | 9.3 | 42.8 | 5.9 | 4.1 | 14.9 | | | |
| *SS20* | | *39.417* | *36.816* | *25* | *13* | *13* | *186.7* | *8.6* | *-31.9* | *13* | *184.5* | *7.6* | *13.5* | *14.5* | *30.9* | *7.6* | *4.3* | *16.3* | | | |
| *SS21* | | *39.419* | *36.803* | *27* | | | | | | | | | | | | | | | | | |
| SS29 | | 39.347 | 36.669 | 17 | 5 | 5 | 179.7 | 12 | -26.1 | 20 | 207.9 | 15 | -32 | 22.7 | 29.5 | 14.3 | 6.3 | 29.7 | | | |
| *SS30* | | *39.280* | *36.607* | *17* | | | | | | | | | | | | | | | | | |
| *SS31 (pdf)* | | *39.451* | *36.952* | *17* | *17* | *17* | *3.6* | *5.9* | *52.4* | *5.3* | *355.6* | *13* | *74.5* | *3.9* | *31.7* | *6.4* | *3.9* | *13.8* | | | |
| **Akkisla** | Eocene | 39.0 | 36.2 | 126 | 87 | 79 | 332.4 | 3.7 | 31.3 | 5.7 | 21.0 | 3.6 | 2.1 | 5.2 | 27.6 | 3.7 | | | | | ccw |
| SS1 | | 39.054 | 36.329 | 21 | 15 | 15 | 210.3 | 13.5 | -64.8 | 7.1 | 165.0 | 6.7 | -38.4 | 8.9 | 38.3 | 6.3 | 4.1 | 14.9 | | | |
| *SS4 (pdf)* | | *38.910* | *36.106* | *10* | *6* | *6* | *1.6* | *14.9* | *55.2* | *11.9* | *27.2* | *13.9* | *51.6* | *12.7* | *33.5* | *11.7* | *5.9* | *26.5* | | | |
| SS5 | | 38.929 | 36.112 | 21 | 19 | 19 | 347.5 | 6.6 | 51.2 | 6.1 | 338.5 | 4.5 | 32.3 | 6.8 | 61.7 | 4.3 | 3.7 | 12.8 | | | |
| SS25 | | 38.757 | 36.162 | 20 | 18 | 18 | 146.7 | 6.8 | -17.5 | 12.6 | 137.2 | 6.7 | -13.2 | 12.8 | 28.1 | 6.6 | 3.8 | 13.3 | | | |
| SS27 | | 39.061 | 36.309 | 17 | 17 | 17 | 0.8 | 3.7 | 55.0 | 3.0 | 341.1 | 2.3 | 36.5 | 3.2 | 265.0 | 2.2 | 3.9 | 13.8 | | | |
| SS55 | | 39.049 | 36.314 | 12 | 8 | 8 | 332.6 | 19.7 | 49.1 | 19.4 | 326.5 | 14.8 | 29.4 | 23.3 | 16.2 | 14.2 | 5.2 | 22.1 | | | |
| *SS56 (pdf)* | | *39.031* | *36.286* | *14* | *8* | | *355* | *10.4* | *44* | *12* | *342* | *7.1* | *23* | *12.3* | *35.9* | *9.4* | *5.2* | *22.1* | | | |
| SS57 | | 39.053 | 36.346 | 11 | 10 | 10 | 336.0 | 12.1 | 33.9 | 17.6 | 342.1 | 27.6 | 65.6 | 13.7 | 8.1 | 18.1 | 4.8 | 19.2 | | | |
| **Gürlevik** | Eocene | 39.6 | 37.7 | 76 | 20 | 19 | | | | | 199.8 | 9.5 | -40.2 | 12.1 | 15.6 | 8.8 | 3.7 | 12.8 | 19.8 | 9.5 | cw |
| SS33 | | 39.605 | 37.712 | 18 | 12 | 11 | 217.0 | 17.0 | -44.3 | 19.4 | 186.2 | 15.4 | -38.4 | 20.2 | 11.2 | 14.2 | 4.6 | 18.1 | | | |
| *SS36* | | *39.627* | *37.798* | *18* | | | | | | | | | | | | | | | | | |
| *SS43 (pdf)* | | *39.617* | *37.733* | *18* | | | | | | | | | | | | | | | | | |
| SS44 | | 39.587 | 37.727 | 12 | 8 | 8 | 205.7 | 5.2 | 6.4 | 10.4 | 207.2 | 7.7 | -41.4 | 9.5 | 62.2 | 7.1 | 5.2 | 22.1 | | | |
| *SS45* | | *39.559* | *37.727* | *10* | | | | | | | | | | | | | | | | | |
| **Güllük** | L.Eo-Oligocene | 39.6 | 37.3 | 69 | 24 | 24 | | | | | 322.4 | 11 | 53.9 | 9.0 | 12.2 | 8.8 | 3.4 | 11.1 | 37.6 | 10.7 | ccw |
| *SS32* | | *39.496* | *37.144* | *18* | *18* | *17* | *244.9* | *7.9* | *-63.2* | *4.5* | *275.1* | *4.5* | *-10.0* | *8.8* | *59.4* | *4.5* | *3.8* | *13.3* | | | |
| SS51 | | 39.593 | 37.278 | 14 | 14 | 12 | 251.9 | 16.8 | -65.4 | 8.6 | 137.3 | 12.7 | -49.9 | 12.3 | 14.2 | 10.9 | 4.2 | 15.6 | | | |
| *SS52* | | *39.567* | *37.336* | *13* | *13* | *13* | *230.2* | *6.3* | *0.0* | *12.6* | *224.9* | *4.5* | *-13.9* | *8.6* | *86.1* | *4.5* | *4.3* | *16.3* | | | |
| *SS53* | | *39.575* | *37.352* | *11* | *11* | *11* | *172.1* | *10.7* | *-39.6* | *13.8* | *99.6* | *13.3* | *-43.7* | *15.4* | *15.4* | *12.0* | *4.6* | *18.1* | | | |
| SS54 | | 39.605 | 37.434 | 13 | 10 | 10 | 328.6 | 8.5 | 8.7 | 16.6 | 331.5 | 19.6 | 59.0 | 13.4 | 11.4 | 14.9 | 4.8 | 19.2 | | | |
| **Akdagmadeni** | Eocene | 39.6 | 36.1 | 74 | 33 | 33 | 345.2 | 7.2 | 54.1 | 6.0 | 350.7 | 5.9 | 26.8 | 9.6 | 20.3 | 5.7 | 3.0 | 9.1 | | | |
| SS14 | | 39.613 | 36.093 | 31 | 22 | 22 | 160.7 | 9.6 | -51.7 | 8.7 | 163.7 | 6.8 | -22 | 11.9 | 22.7 | 6.7 | 3.5 | 11.7 | | | |
| SS15 | | 39.600 | 36.128 | 21 | 11 | 11 | 175.0 | 8.1 | -58.0 | 5.8 | 185.4 | 5.6 | -35 | 7.9 | 76.4 | 5.3 | 4.6 | 18.1 | | | |
| *SS16 (pdf)* | | *39.592* | *36.161* | *22* | | | | | | | | | | | | | | | | | |
| **Sincan** | Oligocene | 39.5 | 37.8 | 16 | 16 | 16 | 304.2 | 9.0 | 47.8 | 9.3 | 277.5 | 6.6 | 25.7 | 11.0 | 22.6 | 6.4 | 4.0 | 14.3 | 55.8 | 9.0 | ccw |
| SS34 | | 39.496 | 37.783 | 16 | 16 | 16 | 304.2 | 9.0 | 47.8 | 9.3 | 277.5 | 6.6 | 25.7 | 11.0 | 22.6 | 6.4 | 4.0 | 14.3 | | | |
| **Yeniköy** | Oligocene | 39.2 | 36.4 | 189 | 93 | 89 | | | | | 317.6 | 3.5 | 41.1 | 4.4 | 22.5 | 3.2 | 2.0 | 4.8 | 42.4 | 3.2 | ccw |
| *SS02* | | *39.109* | *36.249* | *28* | *28* | *28* | *12.2* | *1.7* | *37.5* | *1.5* | *14.9* | *1.1* | *37.5* | *1.5* | *681.4* | *1.0* | *3.2* | *10.0* | | | |
| *SS03 (pdf)* | | *39.147* | *36.318* | *22* | *18* | *18* | *355.6* | *3.6* | *53.7* | *3.1* | *347.3* | *1.9* | *12.2* | *3.7* | *321.5* | *1.9* | *3.8* | *13.3* | | | |
| *SS26 (pdf)* | | *39.105* | *36.153* | *21* | *21* | *21* | *10.1* | *2.9* | *54.6* | *2.4* | *2.3* | *3.3* | *57.5* | *2.4* | *148.6* | *2.6* | *3.6* | *12.0* | | | |
| *SS28 (pdf)* | | *39.139* | *36.301* | *5* | *5* | *5* | *0.3* | *9.6* | *53.0* | *8.3* | *350.7* | *21.9* | *24.1* | *37.3* | *13.8* | *21.3* | *6.3* | *29.7* | | | |
| SS58 | | 39.198 | 36.551 | 5 | 5 | 5 | 314.5 | 11.0 | -9.5 | 21.5 | 311.6 | 14.5 | 35.1 | 20.6 | 32.3 | 13.7 | 6.3 | 29.7 | | | |
| *SS59 (pdf)* | | *39.195* | *36.518* | *6* | *5* | *5* | *6.2* | *8.2* | *59.1* | *5.6* | *43.6* | *8.7* | *61.1* | *5.5* | *141.1* | *6.5* | *6.3* | *29.7* | | | |
| Yeniköy | | 39.200 | 36.500 | 102 | 88 | 84 | 306.2 | 3.1 | 26.2 | 5.1 | 318.0 | 3.7 | 41.5 | 4.5 | 22.5 | 3.4 | 2.0 | 5.0 | | | |




| Site | Age | lat (°) | long (°) | Na | Ni | N45 | in situ D | DDx | I | DIx | tilt corrected D | DDx | I | DIx | K | A95 | A95min | A95max | Rot | DRot | sense |
|---|---|---|---|---|---|---|---|---|---|---|---|---|---|---|---|---|---|---|---|---|---|
| **SIVAS BASIN (cont.)** | | | | | | | | | | | | | | | | | | | | | |
| **Inkonak** | **Upper Oligocene** | **39.3** | **37.1** | **173** | **98** | **87** | | | | | **304.0** | **5.0** | **45.9** | **5.4** | **12.8** | **4.4** | **2.0** | **4.9** | **56.0** | **5.0** | **ccw** |
| Inkonak | | 39.315 | 37.060 | 173 | 98 | 87 | 314.3 | 4.2 | 33.7 | 6.1 | 304.0 | 5.0 | 45.9 | 5.4 | 12.8 | 4.4 | 2.0 | 4.9 | | | |
| **Gemerek** | **Miocene** | **39.200** | **36.000** | **99** | **76** | **71** | | | | | **310.1** | **3.8** | **36.6** | **5.2** | **23.1** | **3.6** | **2.2** | **5.6** | **49.9** | **3.8** | **ccw** |
| Gemerek | | 39.163 | 36.048 | 99 | 76 | 71 | 156.4 | 10.2 | 66.4 | 5.0 | 310.1 | 3.8 | 36.6 | 5.2 | 23.1 | 3.6 | 2.2 | 5.6 | | | |
| **Bünyan** | **Miocene** | **38.1** | **39.9** | **17** | **16** | **16** | | | | | | | | | | | | | | | |
| *SS24 (pdf)* | | *38.806* | *39.934* | *17* | *16* | *16* | *357.2* | *5.5* | *57.9* | *4.0* | *355.9* | *6.5* | *62.9* | *3.8* | *63.9* | *4.6* | *4.0* | *14.3* | | | |
| **Sivas** | **Miocene** | **39.6** | **37.0** | **102** | **59** | **59** | **13.1** | **4.8** | **57.0** | **3.6** | | | | | **25.0** | **3.8** | **2.3** | **6.3** | **16.3** | **4.8** | **cw** |
| SS12* | | 39.596 | 37.036 | 32 | 5 | 5 | 264.8 | 10.0 | -1.8 | 20.1 | 228.5 | 10.8 | -40.6 | 13.6 | 60.0 | 10.0 | 6.3 | 29.7 | | | |
| SS13 | | 39.610 | 37.027 | 20 | 17 | 17 | 178.2 | 4.6 | -53.5 | 3.9 | 169.5 | 3.2 | -37.5 | 4.4 | 139.9 | 3.0 | 3.9 | 13.8 | | | |
| SS22 | | 39.624 | 37.018 | 33 | 33 | 33 | 21.8 | 5.4 | 59.1 | 3.7 | 10.8 | 3.4 | 26.5 | 5.5 | 59.7 | 3.3 | 3.0 | 9.1 | | | |
| SS23 | | 39.603 | 37.032 | 17 | 9 | 9 | 199.5 | 13.9 | -52.5 | 12.2 | 179.1 | 11.9 | -43.6 | 13.8 | 24.1 | 10.7 | 5.0 | 20.5 | | | |
| **Zara** | **Miocene** | **39.7** | **37.8** | **35** | **11** | **11** | | | | | | | | | | | | | | | |
| *SS35* | | *39.718* | *37.753* | *11* | *11* | *11* | *187.5* | *3.6* | *-58.2* | *2.6* | *354.8* | *24* | *-82* | *3.5* | *49* | *6.6* | *4.6* | *18.1* | | | |
| *SS37 (pdf)* | | *39.727* | *37.857* | *11* | | | | | | | | | | | | | | | | | |
| *SS42* | | *39.700* | *37.752* | *13* | | | | | | | | | | | | | | | | | |
| **Kemah** | **Miocene** | **39.6** | **38.9** | **31** | **14** | **14** | **6.7** | **5.4** | **56.7** | **4.1** | | | | | **87.1** | **4.3** | **4.2** | **15.6** | **6.7** | **4.1** | **cw** |
| *SS38* | | *39.656* | *39.020* | *7* | | | | | | | | | | | | | | | | | |
| SS39 | | 39.661 | 39.020 | 12 | 9 | 9 | 188.6 | 8.4 | -57.7 | 6.1 | 179.1 | 4.6 | -33 | 6.8 | 137.7 | 4.4 | 5.0 | 20.5 | | | |
| *SS40* | | *39.612* | *38.829* | *7* | | | | | | | | | | | | | | | | | |
| SS41 | | 39.606 | 38.827 | 5 | 5 | 5 | 183.5 | 4.0 | -54.9 | 3.2 | 136 | 12 | -7.6 | 3.3 | 190.2 | 5.6 | 6.3 | 29.7 | | | |
| **Kalkar** | **Miocene** | **39.2** | **35.9** | **36** | **31** | **27** | | | | | **336.1** | **5.9** | **34.3** | **8.4** | **23.4** | **5.5** | **3.1** | **9.6** | **23.9** | **5.9** | **ccw** |
| Kalkar | | 39.223 | 35.943 | 36 | 31 | 27 | 10.1 | 10.3 | 60.0 | 6.8 | 336.1 | 5.9 | 34.3 | 8.4 | 23.4 | 5.5 | 3.1 | 9.6 | | | |
| **LITERATURE REVIEW** | | | | | | | | | | | | | | | | | | | | | |
| **Cappadocia (2,3,10,11)** | **Late Miocene-Pliocene** | **38.864** | **33.903** | **80** | | **77** | | | | | **353.5** | **3.3** | **50.7** | **3.1** | **33.8** | **2.8** | **2.1** | **5.3** | **6.5** | **3.3** | **ccw** |
| **Kirsehir - AAB (9)** | **Upper Cretaceous** | **38.910** | **33.810** | **248** | | **242** | **331.3** | **2.8** | **49.8** | **2.7** | | | | | **14.8** | **2.4** | **1.3** | **2.6** | **28.7** | **2.8** | **ccw** |
| **Kirsehir - KKB (9)** | **Upper Cretaceous** | **39.590** | **33.430** | **268** | | **266** | **352.8** | **2.6** | **56.3** | **2** | | | | | **17.9** | **2.1** | **1.3** | **2.4** | **7.2** | **2.6** | **ccw** |
| **Kirsehir - AYB (9)** | **Upper Cretaceous** | **39.790** | **34.960** | **66** | | **64** | **18.4** | **6** | **54.1** | **5** | | | | | **13.8** | **5** | **2.3** | **6** | **18.4** | **6** | **cw** |
| **Kepezdag (6)** | **Miocene (16-14 Ma)** | **38.300** | **37.600** | **10** | | **10** | | | | | **334.1** | **20** | **51** | **18.1** | **9.5** | **16.5** | **4.8** | **19.2** | **25.9** | **19.5** | **ccw** |
| **Yamadag (6)** | **Miocene (18-9 Ma)** | **39.000** | **38.100** | **64** | | **61** | | | | | **332** | **6.2** | **50.4** | **5.9** | **12.9** | **5.3** | **2.3** | **6.2** | **28** | **6.2** | **ccw** |
| **Eastern Taurides (4,11)** | **Middle Eocene** | **38.600** | **36.400** | **80** | | **80** | | | | | **329.8** | **3.8** | **45.3** | **4.3** | **22.3** | **3.4** | **2.1** | **5.2** | **30.2** | **3.8** | **ccw** |
| **Eastern Taurides Ecemis (11)** | **Upper Cretaceous** | **37.400** | **34.900** | **158** | | **154** | | | | | **318.5** | **3** | **30.4** | **4.7** | **16.1** | **2.9** | **1.6** | **3.4** | **41.5** | **3** | **ccw** |
| **Pontide orocline E limb (5)** | **Upper Cretaceous** | **41.300** | **35.400** | **121** | | **112** | | | | | **32.1** | **4.1** | **40.5** | **5.1** | **13.7** | **3.7** | **1.8** | **4.2** | **32.1** | **4.1** | **cw** |
| **Pontide orocline centre (5)** | **Upper Cretaceous** | **41.800** | **33.600** | **77** | | **77** | | | | | **353.6** | **3.4** | **36.9** | **4.6** | **26.7** | **3.2** | **2.1** | **5.3** | **6.4** | **3.4** | **ccw** |
| **Pontide orocline W limb (5)** | **Upper Cretaceous** | **44.400** | **32.100** | **448** | | **437** | | | | | **334.6** | **1.4** | **44.1** | **1.6** | **28.6** | **1.3** | **1.1** | **1.8** | **25.4** | **1.4** | **ccw** |
| **Eastern Pontides all (4-5, 12-15)** | **Upper Cretaceous-Eocene** | **40.600** | **38.400** | **161** | | **160** | | | | | **359.7** | **2.7** | **42.8** | **3.2** | **22.4** | **2.4** | **1.6** | **3.4** | **0.3** | **2.7** | **ccw** |

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

**Table 1:** Paleomagnetic results presented in this study and localities from literature review.

*Lat/Long (°)* – latitude/longitude of sites/localities, $N_a$ - number of samples analyzed, $N_i$ - number of samples interpreted, $N_{45}$ number of samples after application of a fixed cut-off (45°), $D$ – declination, $I$ - inclination; $DD_x$ - declination error, $DI_x$ - inclination error, $K$ – Fisher

(1953) precision parameter determined for the mean virtual geomagnetic pole (VGP), $A95$ – Fisher (1953) cone of confidence determined from the mean VGP direction. $A95_{min}$ and $A95_{max}$ represent the confidence envelope of Deenen et al., (2011). If A95 falls within this envelope the distribution likely represents paleosecular variation. *Rot* = amount of rotation relative to North, *DRot*= uncertainty in amount of rotation; cw = clockwise; ccw =c outer-clockwise. All site results given in situ and after tilt correction, for localities directions are given in geographic or tectonic coordinates depending on whether we interpreted the magnetization as primary or secondary (based on

positive or negative field tests). See text for explanation.

**Supplementary Information 1:** Paleomagnetic data files compatible with Paleomagnetism.org (Koymans et al., 2016) used in this paper. Paleomagnetic demagnetization files are provided in a folder with .dir files, and contain demagnetization diagrams and our interpretations, viewable in the interpretation portal of paleomagnetism.org. The folder with .pmag files

contains the statistical parameters of sites and localities discussed in this paper and are provided as separate files for the Ulukışla , Sivas, and Tauride basins. In addition, we provide a file with parametrically sampled literature data, compiled from Baydemir, 1990, Channell et al., 1996, Cinku et al., 2016, Gürsoy et al., 2011, Gürsoy et al. 2003, Hisarli et al., 2016, Kissel et al., 2003, Lefebvre et al., 2013, Lucifora, et al., 2012, Meijers et al., 2010 and references therein, Orbay and



Bayburdi, 1979, Piper et al., 2002, Piper et al., 2012, Piper et al., 2013, Platzman et al., 1998, Saribudak, 1989, Tatar et al., 2000, van der Voo, 1968.

**Supplementary Information 2:** Detailed biostratigraphic constraints obtained from calcareous nannofossils are provided for

5   the Berendi locality in the Central Taurides.