# Peer review of "Paleomagnetic constraints on the timing and distribution of Cenozoic rotations in Central and Eastern Anatolia"

_Solid Earth, 2017_

## Referee Comment (RC1) · Anonymous Referee #1 · 1 Aug 2017

The manuscript "Paleomagnetic constraints on the timing and distribution of Cenozoic rotations in Central and Eastern Anatolia" by GuÌĹrer et al. reports a large number of new paleomagnetic data from two sedimentary basins from Turkey with the aim to reconstruct the tectonic evolution of this complex sector of the circum Mediterranean area.

The manuscript is interesting mainly because it contains a new and large paleomagnetic dataset which could help to understand the rotational history of the different blocks which form this part of Anatolia.

However, in the present form the manuscript is not easy to read and the data pre-

sentation is not clean at all, leaving a deep uncertainty in the possible use of these paleomagnetic data for tectonic interpretation

For all these reasons I recommend to deeply revise the manuscript before to resubmit it for another review procedure.

In particular the following points have to be fixed in the analyses of paleomagnetic data before to discuss their tectonic interpretation.

1) The first point concerns the way the authors calculate the ChRM component. This is a fundamental point for determining paleomagnetic rotations. a) In most of the orthogonal diagrams reported in Fig. 5 the ChRM component is forced to pass through the origin. This has been made even when the isolated component doesn't show a progressive decay toward it. In Fig. 5 this is the case of Alihoca (Fig. 5e,f), AkkÄśsÌğla (Fig. 5bb), Divrigi (Fig. 5dd), GuÌĹruÌĹn (Fig. 5ee), ArdÄścÌğlÄś (Fig. 5i), BekcÌğili (Fig. 5l), Sincan (Fig. 5ii), Eminlik (Fig. 5o), Halkapinar (Fig. 5r), Zara (Fig. 5ll,mm), Hasangazi (Fig. 5t), Postalli (Fig. 5u), Topraktepe (Fig. 5x). In some other cases (e.g. Fig. 5d) the ChRM has been selected in a more correct way and doesn't pass to the origin. It is very hard to understand why the Authors have chosen to force or not to the origin the PCA for the different samples. In case a criteria exists it has to be described in the text, otherwise I recommend to not force the PCA to the origin and to recalculate it for all the samples where it has been made. This point is fundamental and has to be fixed in case of resubmission. b) Among the different criteria used for paleomagnetic data analyses the Authors have to consider also the MAD values obtained for each ChRM and use a selection criteria accordingly (MAD<10?). In some cases (e.g. Fig.5b, 5bb among many others) the orthogonal diagram suggest that the MAD for the selected ChRM is very high. Please check all the data accordingly and discard those with high MAD c) in most of the cases there is no correspondence among the AF or TH demagnetization reported to have been used for calculating ChRM in the text and in Figures. In the ArdÄścÌğlÄś locality in the text it is reported " In most cases, linear decay towards the origin occurred at temperatures up to ∼320°C" whereas in Fig. 5i

the last thermal step is 420° C. This discrepancy is very very frequent in the text and has to be fixed. 2) Calculation of the mean direction for each locality. I disagree with the statistical procedure used to calculate the locality mean direction. My opinion (not negotiable) is that the locality mean has to be calculated using the mean direction for each site and not using all together the single directions obtained from the different sites. The latter method increases the "quality" of the statistical parameter Alfa 95 but overweights the role of sites with a large number of samples. In theory using the method proposed by the authors I could have a single site with 100 samples showing a CCW rotation which has the same weight of nine sites distributed in the basin, each one with 10 samples and with very good statistical parameters, showing CW rotation. If a site has good, reliable, acceptable, statistical parameters (low alpha 95 and high k) it must have the same statistical weight of one site with a larger number of samples. 3) Mean direction in geographic or tectonic coordinates? The criteria used to distinguish sites with a post folding remagnetization is not clear at all. In the ArdÄścÌğlÄś locality, such an example, the rotation is calculated using the results in geographic coordinates because "A95 (2.5) is lower than the A95min (3.4)" and because "the tilt-corrected inclination of ∼30° is considerably lower than that for Eurasia in the Late Cretaceous-Paleogene (∼50°), whereas the geographic inclination of ∼45° is not". Since the first observation is true both for tectonic and geographic coordinates ("The two sites share the same bedding and a fold-test is thus not possible") it seems that the Authors prefer to use the geographic coordinates directions because of the low inclination value in tectonic coordinates. This criteria is not acceptable because it is well known that incli-nation in sediments can be shallow than the expected one due to inclination flattening. The erratic criteria in choosing geographic or tectonic coordinates directions has to be avoided and I strongly suggest to only refer to "classic" field test (Fold and reversal tests) to discriminate between post folding or primary ChRM. 4) Reference direction Rotations are always calculated respect to the north and not to the Eurasia Reference poles, even if in some cases the Authors refer to the Eurasia poles for the inclination "∼30° is considerably lower than that for Eurasia in the Late Cretaceous-Paleogene

($\sim$50°)". This is very confusing for the reader. It is true that Eurasia does not rotate too much during the Tertiary, but I think the Author have to use Eurasia reference pole unless they have a clear reason for not, that must be reported in the text.

I think that all these points have to be fixed before using the paleomagnetic dataset for tectonic interpretation. For this reason I have not reviewed and commented the tectonic interpretation and discussion reported in the manuscript.

Introduction and geological settings are very difficult (sometime impossible) to read. They are plenty of geographic and fault names which are not reported in the figures. Please check that all the names in the text will be present in the figures.

Add a figure with the stratigraphic columns of the two basins which allow to show that they have the same stratigraphic evolution.

---

## Referee Comment (RC2) · Anonymous Referee #2 · 15 Aug 2017

The manuscript buy Gürer et al provides a large new paleomagnetic data set from Cretaceous to Miocene sediments from the UlukÄśşla and Sivas basins in the central Anatolia. The data set convincingly demonstrate about 30° Oligocene-Miocene counterclockwise rotations in the UlukÄśşla basin and the surrounding area. The results are ambiguous from the Sivas Basin. I am not a specialist on paleomagnetism; therefore, my comments will be on the tectonic aspect of the manuscript. However, the paleomagnetic data are precisely given and discussed, and assuming that it is correctly interpreted, the manuscript provides a useful and important contribution to the complex geology of central Anatolia.

[Figure]

My main criticism is to the sections "Introduction" and "Geological Setting" (pages 1 to 5), which are poorly written, exceedingly complex, very difficult to follow, somewhat unrelated to the rest of the manuscript, and contain some errors. For example on page 2 it is stated that the "the Pontides comprise a Paleozoic crystalline basement...". However, there are well developed and thick Paleozoic sedimentary sequences in the Pontides, which can traced for hundreds of kilometers. The error stems in regarding the Pontides as a single tectonic unit. There is also confusion about which two subduction zones is referred to on page 1. More importantly, as the paleomagnetic rotations are Oligocene and Miocene in age, it would be much better to describe and discuss only the Tertiary history of Central Anatolia, rather than dwell on the complexities of subduction zones and Tethyan oceans. This would also increase the impact of the manuscript. I would recommend complete rewrite of this part of the manuscript.

Other comments

1. All structures and localities mentioned in the text should be shown in one of the Figures. I could not locate SarÄśz, Gürün, Malatya, and OvacÄśk faults mentioned on page 5.

2. In the text (page 11) the KÄśzÄślkapÄś locality is described as having 13° ccw rotation, which does not tie up with what is shown in Fig. 6.

3. On page 22, it is written that "the Late Cretaceous and the Eocene, when the Tauride rocks were still connected to the downgoing African plate." In this interval, a Tethyan ocean with a subduction zone (Eastern Mediterranean) was between the African Plate and the Taurides, hence the Taurides were not part of the African Plate.

4. Several of the references in the References list are incomplete, e.g., Granot 2016, Dankers et al., 1978, Barrier and Vrielynck, 2008 Blumenthal 1956....

5. It would be very helpful to provide stratigraphic columns for the UlukÄśşla and Sivas basins

---

## Author Comment (AC1) · 27 Sep 2017

The manuscript "Paleomagnetic constraints on the timing and distribution of Cenozoic rotations in Central and Eastern Anatolia" by Guı̀'Lrer et al. reports a large number of new paleomagnetic data from two sedimentary basins from Turkey with the aim to reconstruct the tectonic evolution of this complex sector of the circum Mediterranean

area. The manuscript is interesting mainly because it contains a new and large paleo-magnetic dataset which could help to understand the rotational history of the different blocks which form this part of Anatolia. However, in the present form the manuscript is not easy to read and the data presentation is not clean at all, leaving a deep uncertainty in the possible use of these paleomagnetic data for tectonic interpretation. For all these reasons I recommend to deeply revise the manuscript before to resubmit it for another review procedure.

In particular the following points have to be fixed in the analyses of paleomagnetic data before to discuss their tectonic interpretation.

1) The first point concerns the way the authors calculate the ChRM component. This is a fundamental point for determining paleomagnetic rotations.

a) In most of the orthogonal diagrams reported in Fig. 5 the ChRM component is forced to pass through the origin. This has been made even when the isolated component doesn't show a progressive decay toward it. In Fig. 5 this is the case of Alihoca (Fig. 5e,f), AkkÄ′ssÌ ËŸ gla (Fig. 5bb), Divrigi (Fig. 5dd), GuÌ′LruÌ′Ln (Fig. 5ee), ArdÄ′scÌ ËŸglÄ′s (Fig. 5i), BekcÌËŸ gili (Fig. 5l), Sincan (Fig. 5ii), Eminlik (Fig. 5o), Halkapinar (Fig. 5r), Zara (Fig. 5ll,mm), Hasangazi (Fig. 5t), Postalli (Fig. 5u), Topraktepe (Fig. 5x). In some other cases (e.g. Fig. 5d) the ChRM has been selected in a more correct way and doesn't pass to the origin. It is very hard to understand why the Authors have chosen to force or not to the origin the PCA for the different samples. In case a criteria exists it has to be described in the text, otherwise I recommend to not force the PCA to the origin and to recalculate it for all the samples where it has been made. This point is fundamental and has to be fixed in case of resubmission.

Response: We agree with the reviewer that in many cases the interpretation in Figure 5 is hard to follow, and seems not always consistent. First, we have completely redone Figure 5 and made it clearer to see what has been done or how diagrams have been interpreted. We do not agree with the reviewer that one always should have a single

'optimum' approach: this does not exist since the PCA method is flawed by its very nature to mirror the vectors with respect to the origin in case of anchoring: this produces a MAD value that cannot be compared to not anchoring. This has recently been made very clear by Heslop and Roberts (2016) in JGR (reference is now in the paper) who have fundamentally improved the decision on anchoring. So, the decision 'to anchor or not to anchor' is not based on a firm statistical footing in the approach of Kirschvink (1980). We have now explained our approach in the text as follows:

In interpreting the demagnetization diagrams, we did not rely on criteria for the maximum angular deviation (MAD, Kirschvink, 1980), because this cannot be justified from a statistical standpoint and depends on anchoring or not anchoring to the origin (Heslop and Roberts, 2016). In almost all cases, anchoring produces an artificially low uncertainty estimation (MAD) compared to an unanchored fit; this is inherent in the method used in the PCA analysis. In our interpretations, common sense and consistency of results dictated whether or not to anchor. Although the criteria to anchor or not to anchor have very recently been placed on a firm statistical footing by Bayesian model selection (Heslop and Roberts, 2016), this has not been implemented (yet) in our software.

We have completely rewritten and revised the section on demagnetization results, and made it more consistent.

b) Among the different criteria used for paleomagnetic data analyses the Authors have to consider also the MAD values obtained for each ChRM and use a selection criteria accordingly (MAD<10?). In some cases (e.g. Fig.5b, 5bb among many others) the orthogonal diagram suggest that the MAD for the selected ChRM is very high. Please check all the data accordingly and discard those with high MAD

Response: As outlined above, we do not use the MAD as a criterion because it is a flawed parameter, depending on anchoring or not. We use common sense and our expertise in recognizing good or reasonable results while highly scattered diagrams

(with high MAD value) are not used. For example, in the figure below: the data points in the TH demag cluster above 210C and not anchoring would give a more or less random component while anchoring gives a consistent direction that agrees with the AF demag. Similarly, not anchoring in the AF example would give a reasonable fit, but we have reason to distrust the higher AF fields: they are not as reliable as the lower fields because of a possible GRM or spurious behaviour due to the possible presence of iron sulphides.

Hence the decision to anchor or not is made on an ad hoc basis and depends on the characteristics of the magnetic carrier, the lithology (organic material, coarse or fine-grained sediments), the nature and behaviour of the demagnetization, etc. Spurious behaviour at higher temperatures due to pyrite, for example, removes useful information at those temperatures, but the information at lower temperatures still gives a good estimate of the ChRM. In summary, we use our knowledge and expertise in addition to consistency of behaviour and results to take a decision on the interpretation.

c) in most of the cases there is no correspondence among the AF or TH demagnetization reported to have been used for calculating ChRM in the text and in Figures. In the ArdÄ′scÌ ËŸglÄ′s locality in the text it is reported " In most cases, linear decay towards the origin occurred at temperatures up to 320C" whereas in Fig. 5i the last thermal step is 420 C. This discrepancy is very very frequent in the text and has to be fixed.

Response: Indeed, the temperature in the figure should have been 320 instead of 420. We have now carefully rechecked all our data and the figures, and we have corrected all such mistakes. If we felt that we had better examples to illustrate demagnetization behaviour, we adapted the examples. Where necessary, we recalculated sites and locality means. This did not change the final rotations by more than one or two degrees.

2) Calculation of the mean direction for each locality. I disagree with the statistical procedure used to calculate the locality mean direction. My opinion (not negotiable) is that the locality mean has to be calculated using the mean direction for each site and

not using all together the single directions obtained from the different sites. The latter method increases the "quality" of the statistical parameter Alfa 95 but overweights the role of sites with a large number of samples. In theory using the method proposed by the authors I could have a single site with 100 samples showing a CCW rotation which has the same weight of nine sites distributed in the basin, each one with 10 samples and with very good statistical parameters, showing CW rotation. If a site has good, reliable, acceptable, statistical parameters (low alpha 95 and high k) it must have the same statistical weight of one site with a larger number of samples.

Response: Here, we do not agree with the reviewer for the following reasons: First of all, if we have sites in a locality (tectonic block or basin) that give different (ccw and cw) rotations, we first test whether combining them leads to a VGP distribution with A95>A95max. If that is not the case, as may happen for two sites with small n, there is no statistical basis to infer that e.g. a (small) declination difference has a tectonic origin, but may instead result from insufficient averaging of PSV. If the combined dataset generates an A95>A95max, there is sufficient reason to study what may be the cause of such discrepancy, e.g. a rotation difference. The approach of the reviewer instead assumes that every site averages PSV 'out', without statistical basis, or taking data distributions and precision into account. Secondly, the notion of low alpha95 and high k being 'acceptable' is erroneous and does not agree with statistical (Fisherian) theory on normal distributions, as for instance discussed in text books, e.g. of Bob Butler or Lisa Tauxe. For large N the alpha95 decreases indeed, as it should: for high dispersion distributions (low k) you need more samples to arrive at a smaller cone of confidence (alpha95). The dispersion k of any given distribution becomes more or less stable (invariant) at sufficient N, and paleomagnetic tradition requires N>7. This is based on a firm statistical footing, see the seminal papers of Fisher, Cox, Creer, etc. Contrary to the Van der Voo (1990) criteria, we have realized that 'acceptable' statistics are fundamentally N dependent (Deenen et al., 2011; see their figure 3). More importantly, we use K, A95 of the VGP distribution which is (largely) circular and close to a normal distribution on a sphere, contrary to a directional distribution which is elongated

depending on latitude. Hence, k and alpha95 do fundamentally not correctly describe such directional distributions (other than directions from lavas, or quickly remagnetized distributions). See the work of Constable & Parker, Johnson and co-workers, and Tauxe and Kent (the TK03.GAD model). We have explained this explicitly now in the text:

In determining the means per locality, we averaged all individual directions of the sites of that locality. We therefore break with paleomagnetic tradition to average the site means per locality, although these are given in Table 1. Site means are unit vectors irrespective of the number of samples per site, and therefore site mean cones of confidence (A95) and dispersion (K) are not propagated. By taking all site directions together, sites with more samples have, naturally, more weight. Since we use the Deenen al. (2011) criteria, this approach is warranted because the range of acceptable A95 is N-dependent, contrary to the traditional criteria (e.g. Van der Voo, 1990; see the discussion and figure 3 in Deenen et al. 2011). A95 should fall within the A95min-A95max envelope which becomes stricter ('narrower') with increasing N. The estimate of dispersion (K) of the distribution, however, is largely independent of N (for N sufficiently large, say N>10) and for increasing N becomes an increasingly better estimate for the true dispersion () of the distribution.

This approach has the advantage that valuable statistical and visible information on the precision and reliability of the total, combined distribution is not lost. The added value of our approach is that we have an unbiased estimate to which extent PSV has been sufficiently sampled (and averaged out, for our purpose of tectonic rotations). Typically, we use a sampling strategy that guarantees a 'sufficient number' of samples averaged over a 'sufficiently long' interval of time. Hence, an average based on a larger number of samples is in principle a better representation of PSV. The added value of our approach is that it tests whether the (VGP) scatter obtained from a site, locality, or region, may be straightforwardly explained by PSV, or whether it is smaller than that (which may indicate remagnetization for instance) or larger (which would require additional sources of scatter, e.g. rotations, very large measuring errors, lightning,

etc.).

Finally, we point out that either approach gives an identical result within error. And since all statistical parameters of every site are provided in the table, every reader is free to recalculate the rotations according to his/her own criteria. This would not modify our conclusions on the tectonics. We prefer to stick to our approach as outlined above, however.

3) Mean direction in geographic or tectonic coordinates? The criteria used to distinguish sites with a post folding remagnetization is not clear at all. In the ArdÄ′scÌ ËŸglÄ′s locality, such an example, the rotation is calculated using the results in geographic coordinates because "A95 (2.5) is lower than the A95min (3.4)" and because "the tilt-corrected inclination of 30 is considerably lower than that for Eurasia in the Late Cretaceous- Paleogene (50), whereas the geographic inclination of 45 is not". Since the first observation is true both for tectonic and geographic coordinates ("The two sites share the same bedding and a fold-test is thus not possible") it seems that the Authors prefer to use the geographic coordinates directions because of the low inclination value in tectonic coordinates. This criteria is not acceptable because it is well known that inclination in sediments can be shallow than the expected one due to inclination flattening. The erratic criteria in choosing geographic or tectonic coordinates directions has to be avoided and I strongly suggest to only refer to "classic" field test (Fold and reversal tests) to discriminate between post folding or primary ChRM.

Response: First, to clarify, we have now clearly indicated in Figs 2 and 3 which directions are from a primary, and which are from a secondary magnetization. Furthermore: where possible, we have conducted fold tests – many of them actually, as shown in Figure 7 - as well as reversal tests where possible (not shown). In cases where the fold test is positive, we conclude that the magnetization is likely primary (pre-tilting). In cases where the fold test is negative, we conclude that the magnetization is likely remagnetized and secondary (post-tilting). We still consider the remagnetized direction to hold information, although that information is more difficult to interpret in absence of

constraints on the exact time of that remagnetization. For instance, the remagnetized direction of the Paleocene Berendi locality shows a small counter-clockwise rotation of 17 degrees. We consider this information valuable, because it shows that the Berendi locality has been part of a counterclockwise rotating domain, although the total amount of rotation that that domain underwent since the Paleocene may have been larger than 17 degrees if remagnetization occurred sometime during the rotation phase. We refrain from simply rejecting sites with a negative fold test since it discards useful information.

Finally, the reviewer seems to have misunderstood the point of the A95min test. If A95 is smaller than A95min, there is less scatter in the site than may be expected from PSV. That suggests that insufficient time has been sampled to represent PSV – despite our sampling strategy - and that the magnetization was therefore likely acquired in a much shorter time span than expected from the thickness (or time interval) of the sedimentary unit that was sampled. This thus may suggest remagnetization. Of course, A95 would be smaller before and after tilt correction in such cases, but that is irrelevant. The observation that A95 is smaller than A95min, plus the observation that the in-situ inclination is close to the expected inclination together suggest that the locality may well be remagnetized and we choose a conservative approach and do not make firm interpretations based on this locality. To clarify, we now write the following:

The two sites share the same bedding and a fold-test is thus not possible. The direction in tectonic coordinates would suggest a vertical axis rotation of $45.2 \pm 2.8°$ cw. We note, however, that A95 (2.7) is lower than the A95min (3.5), indicating undersampling of PSV. This may suggest that the magnetization was acquired in a time period that was too short to fully represent PSV, generally thought to average on a ten to hundred-thousand-year timescale (e.g., Deenen et al., 2011, and references therein). Because both sites were collected from several meters of fine-grained sediments, which likely covers a sufficiently long time interval, such undersampling may indicate remagnetization. We further note that the tilt-corrected inclination of $\sim30°$ is considerably lower than the inclination expected for Eurasia in the Late Cretaceous-

Paleogene (ranging 50-55°, with an error of ±3°). Admittedly, the lower inclination may result from compaction-induced inclination shallowing (by more than 20°), but the observation that the A95 is too low to represent PSV points to remagnetization (and hence no inclination error), we consider that the inclination of 47.7 ± 2.6° in geographic coordinates is close to the expected inclination for the plate against which the Anatolian collage accreted in the Cretaceous to Paleogene, which would be consistent with a post-tilt remagnetization. In any case, the locality would indicate a major clockwise rotation of ∼46° since the Late Cretaceous-Paleogene if not remagnetized, or ∼83° following post-tilt remagnetization (Fig. 6c, Table 1).

To avoid confusion, we have now indicated the post-tilt/remagnetized directions with a different colour in Figures 2 and 3 and in fig. 6.

4) Reference direction. Rotations are always calculated respect to the north and not to the Eurasia Reference poles, even if in some cases the Authors refer to the Eurasia poles for the inclination "30 is considerably lower than that for Eurasia in the Late Cretaceous-Paleogene C3 (50)". This is very confusing for the reader. It is true that Eurasia does not rotate too much during the Tertiary, but I think the Author have to use Eurasia reference pole unless they have a clear reason for not, that must be reported in the text. I think that all these points have to be fixed before using the paleomagnetic dataset for tectonic interpretation. For this reason I have not reviewed and commented the tectonic interpretation and discussion reported in the manuscript.

Response: Our results consistently report the observed declinations (corrected for the IGRF deviation at the locality at the time of sampling), and hence give a rotation with respect to true (geographical) North. The reviewer has a valid point that we should mention that over the time interval covered by the localities that we discuss (Late Cretaceous-Miocene) there is no significant rotation of Eurasia according to the APWP. The fact that we find significant and differential block rotations shows that these rotations must have been accommodated along faults. Our paper also identifies the faults and fault zones that are the best candidates to accommodate those rotations.

We now clarify in the beginning of the discussion: Because Eurasia has not significantly rotated around a vertical axis in this time interval (Torsvik et al., 2012), rotation differences found in our localities must have resulted from regional tectonics, also when results from e.g. Upper Cretaceous rocks are compared with those from Eocene rocks.

So, we prefer to use the data as observed/measured. In a few cases we want to test whether a locality may have suffered from remagnetization, and we compare the observed inclination to the one expected at this location or to the GAD inclination. That appears a straightforward way of analysing to us. We have now clarified this (see the modified text above)

Introduction and geological settings are very difficult (sometime impossible) to read. They are plenty of geographic and fault names which are not reported in the figures. Please check that all the names in the text will be present in the figures.

Response: We have carefully revised the introduction and geological setting and omitted information that is not strictly necessary to understand our analysis and aims. See also the rebuttal to the second reviewer, who had more specific comments on these sections. We have added all geographic names mentioned in the text to the maps of Figures 1-3.

Add a figure with the stratigraphic columns of the two basins which allow to show that they have the same stratigraphic evolution.

Response: Here we refer the reviewer to the existing literature concerned with the stratigraphy of the two basins. For the purpose of this paper we consider only the age of the sedimentary fill and do not see the added value of showing stratigraphic columns. Additionally, particularly the Sivas basin has strong along-strike variations in its stratigraphy, owing to local sub-basins. If we were to follow the reviewer's suggestion, we would have to show multiple stratigraphic columns per basin, which is beyond the scope of this paper, in which we focus on rotation differences through time, irrespective of stratigraphic evolution.
The manuscript buy Gürer et al provides a large new paleomagnetic data set from Cretaceous to Miocene sediments from the UlukÄ′sÂÿsla and Sivas basins in the central Anatolia. The data set convincingly demonstrate about 30 Oligocene-Miocene counterclockwise rotations in the UlukÄ′sÂÿsla basin and the surrounding area. The results are ambiguous from the Sivas Basin. I am not a specialist on paleomagnetism; therefore, my comments will be on the tectonic aspect of the manuscript. However, the paleomagnetic data are precisely given and discussed, and assuming that it is correctly interpreted, the manuscript provides a useful and important contribution to the complex geology of central Anatolia.

My main criticism is to the sections "Introduction" and "Geological Setting" (pages 1 to 5), which are poorly written, exceedingly complex, very difficult to follow, somewhat unrelated to the rest of the manuscript, and contain some errors. For example on page 2 it is stated that the "the Pontides comprise a Paleozoic crystalline basement...". However, there are well developed and thick Paleozoic sedimentary sequences in the Pontides, which can traced for hundreds of kilometers. The error stems in regarding the Pontides as a single tectonic unit. There is also confusion about which two subduction zones is referred to on page 1. More importantly, as the paleomagnetic rotations are Oligocene and Miocene in age, it would be much better to describe and discuss only the Tertiary history of Central Anatolia, rather than dwell on the complexities of subduction zones and Tethyan oceans. This would also increase the impact of the manuscript. I would recommend complete rewrite of this part of the manuscript.

Response: We thank the reviewer for this comment. We have completely rewritten the introduction and the geological setting that described the basement evolution of Central Anatolia. We have shortened both sections considerably, and focus on the aim

of the paper: to identify the coherently rotating domains of Anatolia, and to constrain the timing and amount of rotation, as well as the bounding fault zones. The geological setting was stripped from plate tectonic inferences on Anatolia's subduction history, and merely focuses on providing relative background information required to understand our sampling strategy, and the basic geological architecture of the study area. Details of the Pontide structure and history were omitted as they are irrelevant for our study.

Other comments

1. All structures and localities mentioned in the text should be shown in one of the Figures. I could not locate SarÄ'sz, Gürün, Malatya, and OvacÄ'sk faults mentioned on page 5.

Response: We have carefully revised the figures and have added all geographic names mentioned in the text to the maps of Figures 1-3.

2. In the text (page 11) the KÄ'szÄ'slkapÄ's locality is described as having 13 ccw rotation, which does not tie up with what is shown in Fig. 6. We interpret the Upper Paleocene KÄśzÄślkapÄś sediments to carry a secondary, post-tilt magnetization, which shows a ~45°ccw rotation. The 13 degrees referred to by the reviewer concerns the declination in tectonic coordinates, but the magnetization is not interpreted as primary. We have rewritten the KÄśzÄślkapÄś locality description and clarified this point.

Response: We have clarified further by adding to the discussion section, where Kizilkapi is discussed: This indicates a post-tilt yet still pre-rotation magnetization, but given the inferred secondary nature of the magnetization, we refrain from using this locality in computing the rotation of the UlukÄśşla Basin.

3. On page 22, it is written that "the Late Cretaceous and the Eocene, when the Tauride rocks were still connected to the downgoing African Plate." In this interval, a Tethyan ocean with a subduction zone (Eastern Mediterranean) was between the African Plate and the Taurides, hence the Taurides were not part of the African Plate.

Response: We respectfully disagree with the reviewer. During the Late Cretaceous, subduction indeed occurred between Africa and the Tauride block which led to emplacement of ophiolites onto the Arabian, north African, and south Tauride margin, but this subduction zone is explained and kinematically constrained by ophiolite data (Maffione et al., 2017) by westward roll-back from an originally east-dipping subduction zone that formed east of the Tauride block. This is a similar situation as today's Banda arc. Until the phase of thrusting in Eocene time, the Tauride lithosphere was still connected to Africa, much like the Bird's Head north of the Banda arc is still connected to Australia. We therefore think it is fair to test whether the pre-Eocene rotations of the Taurides may be explained by African rotations.

4. Several of the references in the References list are incomplete, e.g., Granot 2016, Dankers et al., 1978, Barrier and Vrielynck, 2008 Blumenthal 1956....

Response: We have carefully revised the reference list and corrected the inconsistencies pointed out by the reviewer.

5. It would be very helpful to provide stratigraphic columns for the UlukÄ′sÂÿsla and Sivas basins

Response: As we responded to reviewer#1, the stratigraphic columns are not necessary to understand the paleomagnetic and structural analysis of this paper. The stratigraphy merely provides an age for the sampled localities and the age and nature of the sampled rocks are provided in the locality descriptions.

---

## Author Response (AR2)

**Report #1 = Referee #2 (Aral Okay): Accepted as is**

No further comments needed

**Report #2 = Referee #3 (Anonymous): Accepted subject to minor revision**

The manuscript "Paleomagnetic constraints on the timing and distribution of Cenozoic rotations in Central and Eastern Anatolia" presents an impressive amount of new paleomagnetic data for a very interesting and very complex area. I think their approach of obtaining as many paleomagnetic sites as possible represents the only way in dealing with such a complicated tectonic area and definitely deserves publication. It is well written, I think the geological introduction is sufficiently detailed, the structure might need improvement, especially in terms of a summary of the results, but I agree that the topic and results are quite numerous and complex.

On the other hand, however, I think there still remains a problem in the data treatment. My only major concern is the already mentioned treatment of individual sample directions rather than site mean directions. I agree with the authors that always applying the standard paleomagnetic procedures is not always the best thing to do and new approaches are sometimes reasonable and it makes sense to explore those. However, I have the feeling that this area and the presented data might not be the perfect place to introduce a "break with paleomagnetic tradition". In my opinion, using individual directions rather than site mean directions is reasonable when treating with inclination shallowing, or when studying different parts of a continuous section. In such a tectonic complex area, however, local rotations might vary a lot on a small scale, which would require site mean directions. Then, a site with a small number of samples and high alphas (or As) is sometimes very important and equally necessary than another site with high quality results and a large number of samples. To exaggerate, treating 2 separated sites, one with 100 directions, and the other with 10, even if they yield mean directions which are statistically different, would yield a mean direction equal to the first site. The authors talk a bit about that, but I think a much more rigorous introduction of this approach would be necessary to justify it. A way to get around this, would be the presentation of regional mean directions based on both site means and individual means. A small table 2, which shows that the two methods are comparable, but the error is smaller (?) would leave everyone with the choice what to choose. I would say it is necessary to present the standard method at first, and then the alternative. The missing propagation of errors is true, but on one hand, error propagation exists and, I think, this would be a better way to treat this problem. The presented alternative approach does not do propagation either.

Table 1 already contained all site averages and an average of locality means for the Ulukisla basin. The amount of successful sites per locality is generally only 2 or 3, which makes the alpha 95 of the resulting site mean per locality essentially meaningless. Therefore, to accommodate the wishes of the reviewer, we have added

one average for the Ulukisla basin based on site means – leading, as expected and already mentioned in our previous rebuttal and manucscript, to an average rotation that is not statistically different from the averages based on directions (revised Table 1)

In addition, we have added an APWP for SE Anatolia, in which we average sites with a 20 Myr sliding window on 10 Myr intervals, which is the standard approach for APWPs (e.g., Torsvik et al., 2012). This shows that there was a 30 degree ccw rotation in Oligo-Miocene time – as we have concluded before – and that larger deviations of declinations relative to Eurasia for Cretaceous time is straightforwardly explained by motion as part of the African Plate – as we had also concluded before.

One thing, which is also absolutely necessary for the presented method, is a clear discussion about structural data. Which sites have which bedding? So combining individual directions is based on sites with similar bedding? This needs further discussion.

We disagree, this does not really need discussion or additional explanation. Only directions, sites or localities that were concluded to carry a primary magnetization were combined, in tectonic (i.e. tilt-corrected) coordinates. This is standard procedure, regardless of the method used.

In this context, I would like an additional discussion about block rotations versus more continuous deformation? I am not so familiar with the region, but it seems that there is an orocline in the north and one in the south with continuously changing declination (figure 8), but a more blocky behavior in the center. This might well be true, but a small discussion would be nice.

The aim of this paper is to identify rotating domains, their boundaries, and the timing of rotation. The main conclusion in this paper is that SE Anatolia consists of one coherently rotating block, in which internal deformation generally does not lead to significant vertical axis rotations, except where discussed for individual localities. We have described the rotating blocks and their bounding fault zones in sufficient detail. A discussion on continuous versus block deformation does not have any added value, particularly because there is no structural or paleomagnetic evidence for any.

Also the white block boundary in figure 8 might be a bit speculative.

Correct, we have made this a dotted line. The western boundary is poorly constrained as it is buried below recent sediments of the Tuzgölü Basin.

Maybe the red stripes for coherently rotating domains should be restricted to the center? In the end, I think the authors have to decide what they want to do.

The red stripes indicate the general average declination for the Paleogene, and is rather schematic. For full tectonic reconstruction, we build APWPs, as we have now done for the domain where we collected new data, and compare these to predicted declinations from our kinematic restoration (see Li et al., Earth-Sci Rev). This approach is beyond the scope of this paper, were we deliberately choose to show paleomagnetic patterns, not the tectonic evolution that we deduce from that, which will be subject for future work.

Either they present a statement in favor of their alternative statistical approach, which would require more comparisons, a more straight
forward discussion for the two different approaches, and a more simple tectonic setup. Or they tackle the tectonic history of the presented area, which would require at least also to show the site mean directions and do it the classical way.
It is definitely an interesting discussion the authors present here. Also in my opinion the Deenen et al. paper is a major step forward. However, if site mean directions do not fulfil the requirements, using individual directions need to be further justified. I don't think that increasing n and decreasing error is always better.

We have provided the analysis with site mean directions as requested. The tectonic history requires a full review of also structural history, and a kinematic restoration, which is not the purpose of the present study. And as we have rebutted before, increasing N and decreasing error is not relevant in the light of the Deenen et al. method we use: it is N-DEPENDENT, contrary to e.g. the van der Voo criteria.
Hence, with higher N the required A95 becomes stricter (i.e. smaller).

Two more minor points are first, I miss a discussion about inclination shallowing. Because the authors are using arguments based on inclination, some discussion including figures about shallowing would be good. The directions do not seem to show much elongation, but given the amount of results, a site by site inclination shallowing inspection might yield additional information about primary or secondary (if shallowing is present, remagnetization is less likely).

Analysis of inclination shallowing using direction distributions requires N>100 (Tauxe & Kent, 2004) in a single site so as to avoid elongation as a result of small rotation differences between sites. None of our sites fulfill this criterion, and a meaningful discussion of inclination shallowing on a site-by-site basis is thus not possible.
In several cases, where other arguments may suggest that remagnetization may have happened in a site, we compare inclinations in *in situ* and in tectonic coordinates. Where *in situ* inclinations are close to the recent field, AND in tectonic coordinates shallow inclinations arise, we indicate that remagnetization may have happened, and treat the direction more carefully. See also our previous rebuttal where we explicitly and exhaustively rebutted this issue.

Also, it is not easy for the reader to assess the actual outcomes of the study. It would be nice to see some additional summarizing figures, like e.g.   D versus time or a

sketch showing blocks rotating? Most of the rotation arrows in the last figure are already in the first figure. Figure 2 and 3 show all the results (right?).
So, what did change with the presented results? Is there a consistent rotation over time? What is the rotation rate?
The newly added APWP, Figure 9 and Table 2, provide this requested information.

In summary, as I said, I think this is a very interesting data set, which deserves publication after some small revision (most of my comments are rather suggestions and I think not much more modification is needed anymore). I hope my comments help to improve the manuscript.

As the reviewer remarks, the Deenen et al. approach is a major step forward (we agree), and most of the comments above are rather suggestions. The modifications required have now been made.

The paper reports a large number of new paleomagnetic data which are of great interest for interpreting the tectonic evolution of the different Anatolia blocks. However, the data analyses and interpretation are, in my opinion, not adequate and should be completely reconsidered before the paper be acceptable for publication. In the following I briefly discuss my main concern about data analysis.

1) Orthogonal diagrams show in many cases (Fig. 5f, h, o, p, r, s) no-linear path to the origin, whereas the selected magnetic component is fixed to the origin. Furthermore the MAD values, which appear quite large in some diagrams, are never reported in the text or in the figures. In the text the authors report that they interpret the demagnetization diagrams using "common sense and consistency of results", which do not seem a reliable criteria for a scientific paper.

The reviewer points out mostly the same things as the reviewer (then Anonymous Referee #1) in our previous version. We have already explained in the previous rebuttal letter and in the paper that the use of MAD is fundamentally flawed. We see no point in addressing this again, since the reviewer provides no argumentation. We suggest to the editor to make our previous rebuttal available to this reviewer. In the text we already wrote:

'*In interpreting the demagnetization diagrams, we did not rely on criteria for the maximum angular deviation (MAD, Kirschvink, 1980), because this cannot be justified from a statistical standpoint and depends on anchoring or not anchoring to the origin (Heslop and Roberts, 2016). In almost all cases, anchoring produces an artificially low uncertainty estimation (MAD) compared to an unanchored fit; this is inherent in the method used in the PCA analysis. In our interpretations, common sense and consistency of results dictated whether or not to anchor. Although the criteria to anchor or not to anchor have very recently been placed on a firm statistical footing by Bayesian model selection (Heslop and Roberts, 2016), this has not been implemented (yet) in our software.*'

The diagrams in Figures 5 h, o, p, r, and s all converge towards the origin, but are scattered. The reason that these diagrams do not reach the origin is that at higher temperature or coercivity steps, demagnetization behavior becomes erratic, or may be subject to gyroremanence in the case of AF demagnetization. These steps are not shown. Not forcing these demagnetization diagrams to the origin does not make a significant difference. Diagram 5f shows an example where the AF demagnetization does not yield an identical results as the TH demagnetization in 5e. But subsequent TH demagnetization after AF demag does.

Paleomagnetic results are often scattered and do not follow the desired straight lines to the origin of Zijderveld diagrams. As a result, these diagrams should be interpreted with expertise on the type of lithology, magnetomineralogy, etc..

We provide the raw data, so if the reviewer wants to, he or she will be able to make all interpretations him or herself, and find that reinterpretation will make no significant difference. We reiterate our previous rebuttal on this point:

*As outlined above, we do not use the MAD as a criterium because it is a flawed parameter, depending on anchoring or not. We use common sense and our expertise in recognizing good or reasonable results while highy scattered diagrams (with high MAD value) are not used. For example in the figure below: the datapoints in the TH demag cluster above 210C and not anchoring would give a more or less random component while anchoring gives a consistent direction that agrees with the AF demag. Similarly, not anchoring in the AF example would give a reasonable fit, but we have reason to distrust the higher AF fields: they are not as reliable as the lower fields because of a possible GRM or spurious behaviour due to the possible presence of iron sulphides.*

[Figure]

*Hence the decision to anchor or not is made on an ad hoc basis and depends on the characteristics of the magnetic carrier, the lithology (organic material, coarse or fine grained sediments), the nature and behaviour of the demagnetization, etc. Spurious behaviour at higher temperatures due to pyrite, for example, removes useful information at those temperatures, but the information at lower temperatures still gives a good estimate of the ChRM. In summary, we use our knowledge and expertise in addition to consistency of behaviour and results to take a decision on the interpretation.*

2) The calculation of the mean direction for each locality is done using the approach suggested by Deenen et al., 2011, which averages all the directions obtained in each

locality and not the mean direction for each site. I completely disagree with this approach. In fact, as correctly stated by the Authors, in this method the
weight of each site depends from the number of samples. As a consequence a site with 100 samples weights 10 times more than a site with 10 samples. If this is the approach I do not understand why, for each locality, the author decided to sample many sites with few samples instead to concentrate their sampling in a single
site with a large number of samples. For example, in Aktoprak locality they sampled several sites with about 20 samples each, but the locality mean direction was in any case pre determined by the results obtained by the Mejiers et al section, which contains 313 directions. I completely miss the utility of this approach (except
to apparently increase the statistical quality of the data). The use of standard approach in averaging site mean directions is essentially due to the fact that giving the same weight to each site of a locality, independently from the number of samples, we avoid the risk that the presence of one site with a very large
number of samples pre determine the locality mean direction. The classical approach (to distribute a large number of sites, each one with a sufficient number of samples to avoid PSV concerns, in a wide area) allows to have a locality mean direction which be really representative of the locality (or basin) and not of a single site simply because we collected more samples from it.

As explained previously, we hold a strongly different view on this, as we have extensively explained in the previous response to reviewers, as well as in the text of the manuscript. But to accommodate the view of this reviewer, and reviewer #2, we have added two additional calculations to the paper. The first concerns a calculation of the average rotation of the Ulukisla basin using site averages, added to Table 1, in the style desired by the reviewer. The second concerns the calculation of an APWP for the counterclockwise rotating domain of SE Anatolia based on all published sites (so on site averages), with a 20 Myr sliding window, for time intervals where sufficient (>5) sites were available. Those analyses demonstrate that the conclusions we drew in the previous manuscript (32.3±.2.2° ccw rotation) are statistically identical to the desired approach of the reviewer (29.6±4.8° ccw rotation), and the style of calculation makes no difference, other than that in our approach the statistical properties of a locality average can be used to compare to expected scatters from PSV, and thus serve as a reliability criterion. We must again mention that the Deenen et al. approach is N-dependent. See for details our response above, and our previous rebuttal.

3) I disagree to use a 45° cut-off for calculating the site mean direction. In most of the sites all the directions are within the 45°, but in some of them the cut-off eliminate more than 60% of the obtained direction. I suspect that the directions obtained from these sites are not really reliable.

We presume the reviewer refers to site KG2, the only site for which a large difference between pre and post-cutoff *n* was mentioned in Table 1. There, we indicated that 18 out of 30 directions were eliminated by the 45 cutoff. This concerned a typo. 18 cores yielded no interpretable results. Of the 12 interpreted

samples, none were eliminated by the 45 degree cutoff. In all other cases, the amount of directions eliminated by the cutoff  was nil or <10%. The use of the cut-off is a common procedure – it is explained and appropriate references are given in the text.

4) I completely miss the criteria used to define if a site (locality) is magnetically overprinted or not. I totally disagree that a low A95 value be a criteria to assume that the site is remagnetized, unless the authors be very aware of the remagnetization mechanism, which does not seem the case. Magnetic overprint is a very complicated, and not fully understood, process which not necessary occurs in a time span not sufficient to average the PSV. This assumption seems also at odd with the fact that in some localities (Sivas, Kemah) or areas (Tauride) the remagnetized sites show both normal and reverse polarities which suggest that magnetic
overprint occurred during a long time, which should be sufficient to average the PSV. I also disagree that a low inclination be a criteria to decide if a site (locality) be magnetically overprinted. Inclination flattening is a very known process, which can easily explain the low inclination values measured in some sites.

The reviewer misunderstood what the text says. We state that finding a very low A95 may be an indication of remagnetization. Nowhere do we say that sites with high A95 cannot be remagnetized. All we say is that in such cases, A95 values do not provide an argument to infer remagnetization.

Low inclination as such and in isolation is not a hard criterion to infer remagnetization, but in combination with other observations, such as data scatter, regional inconsistency, demagnetization behavior, etc etc, it provides arguments that may lead us to interpret a site with care. Moreover, we explicitly use the combination of an inclination that in *in situ* coordinates is close to the Recent field AND a shallow inclination in tectonic coordinates as argument, as clearly formulated in the text.

In our interpretation section, we carefully assess each locality and discuss whether we are confident in the result, or whether we find arguments that may suggest a non-primary origin of the magnetization. In case of the latter, we choose not to use the locality or site as hard evidence for the rotation history.
Indeed, inclination flattening is a common (but not really well-known) process, but flattening of more than 20° is suspicious in all common lithologies, while together with a GAD field direction before tilt correction makes it *very* suspicious. As we have mentioned, we use common sense and our expertise rather than 'hard rules' to determine whether a site is reliable or not.

In all cases where possible, we have applied fold tests and reversal tests, i.e. the classical tools. Those classical tools simply cannot be used in all localities, because not all are folded and not all have reversals. The 'classical' approach is then to perform a few fold tests and assume that magnetizations in all regions where no fold test was possible are also primary. We prefer not to do so, but apply a more

conservative and nuanced approach whereby we judge each locality based on the available information.

5) I think that the Author can produce a much better tectonic model than that reported in Fig. 8, which does not give any significant information about the tectonic evolution of the area. I think that the approach used to analyse the data is substantially wrong and hide the strong scientific value that this large set of new paleomagnetic data could have for interpreting the tectonic evolution of the area. Before the paper be considered for publication in Solid Earth I strongly suggest to the Authors to use a classical approach to analyse the data (using MAD with a well defined threshold value and not "common sense"; averaging site mean directions and not samples directions for each locality; do not use the 45° cutoff; to use classical field methods to define magnetic overprint instead of strange and unreliable criteria).

The purpose of this paper is explicitly not to develop a tectonic model, and nowhere do we indicate we want to do so. The purpose of this paper is to determine the size or rotating domains, the timing of rotations, and the amount of rotation. Kinematic restoration requires a thorough review of structural history on top of the paleomagnetic evidence, and is the subject for a future paper.

We have already outlined why applying MAD is flawed, and the reviewer provides no counterargument. Classically, a MAD < 15° is used, which is always never reached in case of forcing the interpreted direction through the origin, so applying MAD makes no difference for our result.

We now also calculated the average rotation based on site averages. It makes no significant difference . In addition, we added an APWP for the study area based on all published sites, which confirms the ~30° ccw rotation we concluded before.

We have performed all classical field methods to define magnetic overprint, and used those as conclusive evidence. Our additional arguments only come into play where classical field methods cannot be used. Nowhere does the reviewer indicate how an alternative interpretation may apply, or where the results of a site or locality may be flawed.

Please report the name of each site in Fig. 2 and Fig. 3.
There is insufficient space on these figures to list all site numbers. The coordinates of all site numbers is provided in Table 1.

In conclusion:

The reviewer provides rather strongly worded opinions on how our paper and analysis would be flawed, but none of these arguments have any influence on the conclusions presented in this paper. Referee #3 explicitly mentions that the Deenen et al. paper has been a major step forward in treating paleomagnetic data. Finally, we have complied with the reviewer's request to average site averages by developing an independent, site average-based, APWP for SE Anatolia.

[revised manuscript text omitted]